# Addressing the speed-accuracy simulation trade-off for adaptive spiking neurons

**Luke Taylor**
Department of Physiology, Anatomy and Genetics
University of Oxford
Oxford, United Kingdom
luke.taylor@hertford.ox.ac.uk

**Andrew J King**
Department of Physiology, Anatomy and Genetics
University of Oxford
Oxford, United Kingdom
andrew.king@dpag.ox.ac.uk

**Nicol S Harper**
Department of Physiology, Anatomy and Genetics
University of Oxford
Oxford, United Kingdom
nicol.harper@dpag.ox.ac.uk

## Abstract

The adaptive leaky integrate-and-fire (ALIF) model is fundamental within computational neuroscience and has been instrumental in studying our brains *in silico*. Due to the sequential nature of simulating these neural models, a commonly faced issue is the speed-accuracy trade-off: either accurately simulate a neuron using a small discretisation time-step (DT), which is slow, or more quickly simulate a neuron using a larger DT and incur a loss in simulation accuracy. Here we provide a solution to this dilemma, by algorithmically reinterpreting the ALIF model, reducing the sequential simulation complexity and permitting a more efficient parallelisation on GPUs. We computationally validate our implementation to obtain over a $50\times$ training speedup using small DTs on synthetic benchmarks. We also obtained a comparable performance to the standard ALIF implementation on different supervised classification tasks - yet in a fraction of the training time. Lastly, we showcase how our model makes it possible to quickly and accurately fit real electrophysiological recordings of cortical neurons, where very fine sub-millisecond DTs are crucial for capturing exact spike timing.

## 1 Introduction

The surge of progress in artificial neural networks (ANNs) over the last decade has advanced our understanding of the potential computational principles underlying the processing of sensory information [1–9]. Although these networks architecturally bear a resemblance to the brain [10], they tend to omit a key physiological constraint: the spike. With their increased biological realism, spiking neural networks (SNNs) have shown great promise in bridging the gap between experimental data and computational models. SNNs can be fitted to real neural data [11–15], or used to simulate the

37th Conference on Neural Information Processing Systems (NeurIPS 2023).

brain, offering a new level of understanding of the complex workings of the nervous system [16–19]. They also have engineering applications in energy-efficient machine learning [20].

An established class of spiking models in computational neuroscience is the leaky integrate-and-fire (LIF) neuron, with origins dating back to 1907 [21]. Just like in real neurons, input current (resulting from presynaptic input) charges the membrane potential of the model neurons, which then output binary signals (*i.e.* spikes) as a form of communication (Figure 1a). The adaptive leaky integrate-and-fire (ALIF) model [22] is a modern extension of the LIF. It more closely mimics the biology, capturing a key property of neurons, which is their adaptive firing threshold (*i.e.* spikes become less frequent in response to a steady input current [23]). ALIF neurons have been shown to accurately fit real neural recordings [11, 22, 24, 25] and to outperform the simpler LIF neurons on various machine learning benchmarks [26–29].

Despite these modelling advances, a major shortcoming of LIF and ALIF neurons is their slow inference and training times. Unlike real neurons, modelling neuron dynamics involves sequential computation over discretised time. This leads to a problematic trade-off between speed and accuracy when simulating SNNs. A small DT enables accurate modelling of dynamics, but is slow to stimulate and train on computer systems such as GPUs. A large DT obtains less accurate dynamics, but at the benefit of being faster to simulate and train [30] (Figure 1b). This raises the important question of capturing the best of both worlds: **is there a way to accelerate the inference and training of spiking LIF and ALIF neurons without sacrificing simulation accuracy?**

In this work, we address the speed-accuracy trade-off when simulating and training LIF and ALIF SNNs, and present a solution that is both fast and accurate. We take advantage of a fundamental property of neurons that is sometimes not modelled - the absolute refractory period (ARP). This is a short period of time following spike initiation, during which a neuron cannot fire again. As a result, a neuron can spike at most once within such a period (Figure 1c). We leverage this observation to develop a novel algorithmic reformulation of the ALIF model which reduces the sequential simulation complexity of the standard ALIF algorithm. Specifically, we outline how ALIF recurrent networks can be simulated with a constant sequential complexity $O(1)$ over the ARP simulation length $T_R$, and how this approach can be extended to longer simulation lengths to obtain identical dynamics to the standard ALIF network algorithm. Faster simulation and training are theoretically obtained for growing length $T_R$, by either employing physiologically plausible ARPs of $\sim 2$ms and decreasing the DT to a very fine value $\sim 0.1$ms (for realistic neural modelling), or setting the DT to a coarser value and adopting larger non-physiological ARPs (for machine learning tasks). Our main contributions are the following:

- We develop a novel algorithmic reformulation of the ALIF model with an $O(T/T_R)$ - rather than $O(T)$ - sequential complexity, for simulation length $T$ and ARP length $T_R$.[1]

- We find that our model achieves substantial inference (up to $40\times$) and training speedup (up to $53\times$) over the standard ALIF SNN for increasing ARP time steps, and find this to hold over different numbers of simulation lengths, batch sizes, number of neurons and layers.

- We demonstrate the feasibility of our model to be trained using surrogate gradient descent, with our accelerated ALIF SNN achieving comparable accuracies to the standard ALIF SNN on temporal spiking classification datasets - yet in a fraction of the training time.

- Finally, we showcase how our ALIF implementation makes it possible to quickly fit real electrophysiological recordings of cortical neurons, where very fine sub-millisecond discretisation steps are important for accurately capturing exact spike timing.

---

[1]To avoid ambiguity, simulation length $T$ and ARP length $T_R$ are number of time steps (dimensionless) and not unit time.

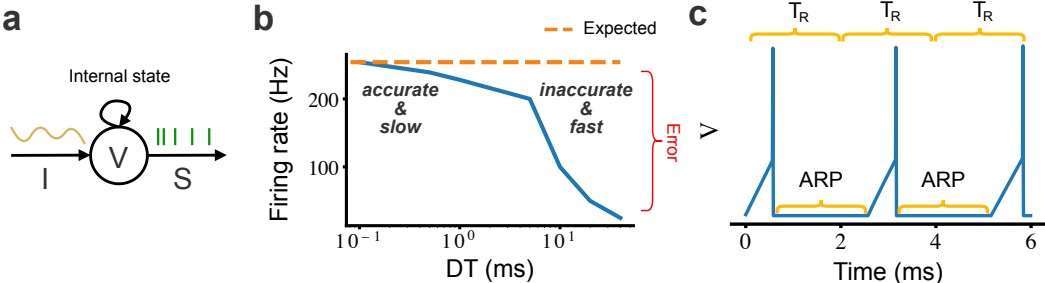

Figure 1: **Problem overview. a.** Schematic of an ALIF neuron: input current $I$ charges membrane potential $V$ and outputs spikes $S$ if firing threshold is reached (with the neuron's internal state evolving over time). **b.** An example of the simulation trade-off problem when simulating a single ALIF neuron with fixed synaptic weights receiving Poisson spike input. The simulation error and the speed grow for increasing discretisation time (DT). **c.** Observation for our solution: a neuron emits at most a single spike during a simulation span $T_R$ equal in length to the neuron's absolute refractory period (ARP).

## 2 Background and related work

**Standard ALIF model** We introduce the recurrent SNN of ALIF neurons with fixed ARP and latency of recurrent transmission, defined by the following set of equations [26, 31].

$$S_i^{(l)}[t] = f(V_i^{(l)}[t]) = \mathbb{1}_{V_i^{(l)}[t] > \theta_i^{(l)}[t]} \quad \text{(Output spike)} \tag{1}$$

$$V_i^{(l)}[t] = \big(\beta_i^{(l)} V_i^{(l)}[t-1] + (1 - \beta_i^{(l)}) I_i^{(l)}[t]\big)\big(1 - S_i^{(l)}[t-1]\big) \quad \text{(Membrane potential)} \tag{2}$$

$$\tilde{I}_i^{(l)}[t] = \Big(b_i^{(l)} + \underbrace{\sum_{j=1}^{N^{(l-1)}} W_{ij}^{(l)} S_j^{(l-1)}[t]}_{\text{Feedforward current}} + \underbrace{\sum_{j=1}^{N^{(l)}} W_{ij}^{\text{rec}\,(l)} S_j^{(l)}[t-D]}_{\text{Recurrent current}}\Big) \quad \text{(Input current)} \tag{3}$$

$$I_i^{(l)}[t] = \begin{cases} \tilde{I}_i^{(l)}[t] & \text{if } C_i^{(l)} \geq T_R \\ 0 & \text{otherwise} \end{cases} \quad \text{(Absolute refractory period)} \tag{4}$$

$$\begin{aligned} \theta_i^{(l)}[t] &= 1 + d_i^{(l)} a_i^{(l)}[t] \\ a_i^{(l)}[t] &= p_i^{(l)} a_i^{(l)}[t-1] + S_i^{(l)}[t-1] \end{aligned} \quad \text{(Adaptation)} \tag{5}$$

At time $t$, neuron $i$ within layer $l$ (consisting of $N^{(l)}$ neurons) receives input current $I_i^{(l)}[t]$ and outputs a binary value $S_i^{(l)}[t] \in \{1, 0\}$ (*i.e.* spike or no spike) if a neuron's membrane potential $V_i^{(l)}[t]$ reaches firing threshold $\theta_i^{(l)}$ (Equation 1).[2] The evolution of the membrane potential is described by the normalised discretised LIF equation (Equation 2), in which the membrane potential dissipates by a learnable factor $0 \leq \beta_i^{(l)} \leq 1$ at every time step and resets to zero if a spike occurred at the previous time step. The input current is comprised of a constant bias source $b_i^{(l)}$ and from incoming spikes reflecting feedforward $W^{(l)} \in \mathbb{R}^{N^{(l)} \times N^{(l-1)}}$ and recurrent connectivity $W^{\text{rec}\,(l)} \in \mathbb{R}^{N^{(l)} \times N^{(l)}}$ (Equation 3).

Here, we assume the recurrent transmission latencies $D$ to be of fixed length and equal in length to the ARP, that is $T_R = D$. With a simple modification however, our methods can work for longer $D$, or different $D$ on each connection, so long as $D \geq T_R$. The ARP is enforced by only allowing input current to enter the neuron if the number of time steps $C_i^{(l)}$ following the last spike equals or exceeds the ARP length (Equation 4). Lastly, adaptation is implemented by raising the firing threshold $\theta_i^{(l)}[t]$

---

[2]Here notation $\mathbb{1}_{\text{condition}}$ denotes the indicator function, which is equal to one if the condition is true and zero otherwise.

following each spike $S_i^{(l)}[t-1]$ (Equation 5), which decays exponentially to baseline $\theta_i^{(l)} = 1$ in the absence of any spikes (using learnt decay factor $0 \leq p_i^{(l)} \leq 1$ and adaptation scalar $d_i^{(l)}$).

The ARP in biological neurons is typically about $\sim 1 - 2$ms [32–34]. Our method takes advantage of the fact that the monosynaptic connection latency between neurons in local circuits is typically also often around $\sim 1 - 2$ms [35]. LIF and ALIF neuronal models are typically run with a DT of about $\sim 0.1$ms in the computational neuroscience literature and $\sim 1$ms in machine learning literature [30]. Furthermore, the firing rate of neurons is often less than the reciprocal of the ARP, suggesting that a higher $T_R$ than the ARP may still provide a reasonable approximation of neural behaviour for some purposes.

**SNN training** The main problem with training SNNs is the non-differentiable nature of their activation function in Equation 1 (whose derivative is undefined at $V_i^{(l)}[t] = \theta_i^{(l)}[t]$ and zero otherwise). This precludes the direct use of the backprop algorithm [36], which has underpinned the successful training of ANNs. A popular solution is to replace the undefined spike derivative with a well-behaved function, referred to as a surrogate gradient [37–40], and training the network with backprop-through-time [41]. This method also supports the training of neural parameters other than synaptic connectivity, such as membrane time constants [42, 28] and adaptive firing thresholds [26–28], both of which we include in our work, as they have been shown to improve performance (and increase biological realism). It is worth mentioning that other SNN training methods exist such as mapping trained ANNs to SNNs by transforming ANN neuron activations to SNN neuron firing rates [43–46]. These methods are, however, of less interest to computational neuroscientists as they discard all temporal spike precision.

**Related work** Recent work has provided new theoretical insights for increasing the simulation accuracy when employing a larger DT $\geq 1$ms [30]. We are not aware of any work that accelerates the simulation and training times on GPUs when employing a smaller DT $\leq 1$ms (although see the NEST simulation library for simulating SNNs on CPUs [47–49]). There are, however, different methods for speeding up the backward pass (*i.e.* how gradients are computed within the SNN), which consequentially speeds up training times. Rather than being viewed as competing approaches, these methods could further augment the speed of our solution. In sparse gradient descent, gradients are only passed through neurons whose membrane potential is close to firing threshold, which can significantly accelerate the backward pass when neurons are mostly silent [50]. Inspired by work on training non-spiking networks [51, 52], other methods completely bypass the backward pass and adjust weights online [53, 27, 29]. Another approach is to propagate gradient information through spikes [54–60], which - similar to the idea of sparse gradient descent - is fast when neurons are mostly silent. This method, however, enforces neurons to spike at most once and can suffer from training instabilities (although see [61, 62]).

## 3 Theoretical results

We outline a novel reformulation of the ALIF model, which theoretically reduces the sequential simulation complexity from $O(T)$ to $O(T/T_R)$ (for simulation length $T$ and ARP length $T_R$). Using the observation that a neuron can spike at most once over simulation length $T_R$ (equal to the ARP; Figure 1c), we propose simulating network dynamics sequentially in blocks of length $T_R$ - as opposed to simulating every time step individually (Figure 2a) - as we show that these blocks can be simulated with a constant sequential complexity $O(1)$. Our reformulated ALIF model is mathematically the same as the standard ALIF model, but substantially faster to simulate and train. All the proofs for the propositions can be found in the Supplementary material.

### 3.1 Block: simulating single-spike ALIF dynamics with a constant sequential complexity

A SNN exhibits a sequential dependence due to the spike reset mechanism, where a neuron's membrane potential $V_i[t]$ can be reset based on its previous output $S_i[t-1]$. Consequently, the simulation of a SNN necessitates a sequential approach. This restriction can, however, be alleviated if we assume a neuron to spike at most once over a particular simulation length (such as the ARP). The following steps - grouped together into a module referred to as a Block (Figure 2b) - compute

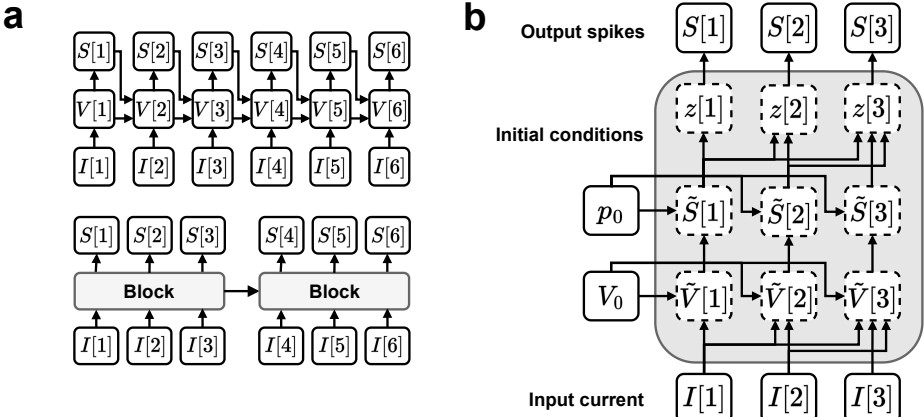

Figure 2: **Solution overview**. **a.** Our proposed solution: instead of simulating network dynamics one time step after another (top), we sequentially simulate blocks of time equal in length to the neuron ARP (bottom), in which a neuron can spike at most once. **b.** A schematic of a Block: our proposed solution for emulating ALIF dynamics with a constant sequential complexity $O(1)$ over a short duration in which a neuron spikes at most once.

ALIF dynamics (assuming at most one spike) without any sequential operations.

$$\tilde{V}_i[t] = \left(I_i * \tilde{\beta}_i\right)[t] \quad \text{(No-reset membrane potential)} \tag{6}$$

$$\tilde{S}_i[t] = f(\tilde{V}_i[t]) \quad \text{(Faulty output spikes)} \tag{7}$$

$$z_i[t] = \phi(\tilde{S}_i)[t] \quad \text{(Latent timing of spikes)} \tag{8}$$

$$S_i[t] = \mathbb{1}_{z_i[t] = 1} \quad \text{(Correct output spikes)} \tag{9}$$

**1. Calculate membrane potentials without reset**    The first step in the Block (Equation 6) converts input current $I_i[t]$ to membrane potentials $\tilde{V}_i[t]$ without spike reset (*i.e.* excluding the reset mechanism in Equation 2). This transformation is achieved using a convolution (Proposition 1), thus avoiding any sequential operations.

**Proposition 1.** *Membrane potentials without spike reset are computed as a convolution* $\tilde{V}_i[t] = \left(I_i * \tilde{\beta}_i\right)[t]$ *between input current $I_i[t]$ and kernel $\tilde{\beta}_i[t] = (1 - \beta_i)\beta_i^t$ with the initial membrane potential encoded as $I_i[0] = \frac{V_i[0]}{1-\beta_i}$.*

**2. Faulty output spikes**    No-reset membrane potentials $\tilde{V}_i[t]$ are mapped to erroneous output spikes $\tilde{S}_i[t] = f(\tilde{V}_i[t])$ (Equation 7) using spike function $f$ (Equation 1). This output can contain more than one spike, but only the first spike complies with the standard model dynamics, due to the omission of the reset mechanism and ARP constraint. Thus, to ensure that the Block only emits a single spike, all spikes succeeding the first spike occurrence are removed using the following steps.

**3. Latent timing of spikes**    Erroneous spike output $\tilde{S}_i[t]$ is mapped to a latent representation $z_i[t] = \phi(\tilde{S}_i)[t]$ (Equation 8), encoding the timing of spikes. Function $\phi(\cdot)$ (taking vector $\tilde{S}_i$ as input; Proposition 2) is constructed to map all erroneous spikes $\tilde{S}_i[t]$, besides the first spike occurrence, to a value other than one (*i.e.* $z_i[t] \neq 1$ for all $t$ except for the smallest $t$ satisfying $\tilde{S}_i[t] = 1$ if such $t$ exists).

**Proposition 2.** *Function $\phi(\tilde{S}_i)[t] = \sum_{k=1}^{t} \tilde{S}_i[k](t - k + 1)$ acting on $\tilde{S}_i \in \{0, 1\}^T$ contains at most one element equal to one $\phi(\tilde{S}_i)[t] = 1$ for the smallest $t$ satisfying $\tilde{S}_i[t] = 1$ (if such $t$ exists).*

**4. Correct output spikes**    Lastly, the correct spike output $S_i[t] = \mathbb{1}_{z_i[t] = 1}$ is obtained by setting every value in $z_i[t]$, besides the value one (*i.e.* the first spike), to zero (Equation 9).

## 3.2 Blocks: simulating ALIF SNN dynamics with a $O(T/T_R)$ sequential complexity

The standard ALIF SNN model can be reformulated using a chaining of Blocks, which reduces the sequential simulation complexity (as each Block is simulated in $O(1)$). For a given ARP of length $T_R$, we observed that a neuron spikes at most once over simulation length $T_R$ (Figure 1c). Thus, a simulation length $T$ can be simulated using $N = \frac{T}{T_R}$ Blocks, each of length $T_R$. [3] Next, we outline how to simulate ALIF dynamics across Blocks to emulate the dynamics of the standard ALIF SNN. We introduce new notation to index Block $1 \leq n \leq N$ using subscript $n$ (*e.g.* input current to neuron $i$ simulated in Block $n$ is expressed as $I_{i,n}[t]$) with time steps indexed between $[1, T_R]$ (as opposed to $[1, T]$) within a Block.

**Input current and the ARP of a Block**  The input current in the standard model (Equation 3 and 4) is modified for the Block model (Proposition 3). The feedforward and recurrent current to Block $n+1$ are derived from the presynaptic and postsynaptic spikes from Block $n + 1$ and Block $n$ respectively. In addition, the ARP is enforced by applying a mask derived from the latent timing of spikes $z_{i,n}[t]$ (Equation 8).

**Proposition 3.** *The input current $I_{i,n+1}[t]$ of neuron $i$ simulated in Block $n + 1$ (of length $T_R$) is defined as follows, and enforces an absolute refractory period of length $T_R$ and a monosynaptic transmission latency of $D = T_R$.*

$$I_{i,n+1}[t] = \Big( \underbrace{b_i + \sum_{j=1}^{N^{in}} W_{ij} S_{j,n+1}[t]}_{\text{Feedforward current}} + \underbrace{\sum_{j=1}^{N^{out}} W_{ij}^{rec} S_{j,n}[t]}_{\text{Recurrent current}} \Big) \underbrace{\mathbb{1}_{z_{i,n}[t] \geq \max_t S_{i,n}[t]}}_{\text{ARP mask}}$$

**Evolving membrane potentials between Blocks**  Two cases are distinguished to correctly emulate the evolution of the membrane potentials between Blocks (Proposition 4): 1) if neuron $i$ did not spike in Block $n$ (*i.e.* $\max_t S_{i,n}[t] = 0$), then its initial membrane potential $V_{i,n+1}[0]$ in Block $n + 1$ is set to its final membrane potential $V_{i,n}[T_R]$ in Block $n$. Otherwise 2), the initial membrane potential is set to zero to emulate a spike reset (and no state needs to be transferred between Blocks as the neuron is in a refractory state).

**Proposition 4.** *The initial membrane potential $V_{i,n+1}[0]$ of neuron $i$ simulated in Block $n + 1$ (of length $T_R$) is equal to the last membrane potential in Block $n$ if no spike occurred and zero otherwise.*

$$V_{i,n+1}[0] = \begin{cases} V_{i,n}[T_R] & \text{if } \max_t S_{i,n}[t] = 0 \\ 0 & \text{otherwise} \end{cases}$$

**Evolving adaptive firing thresholds between Blocks**  The adaptive firing threshold $\theta_{i,n+1}[t]$ of neuron $i$ in Block $n + 1$ is derived from the initial adaptive parameter $a_{i,n+1}[0]$ (Proposition 5). Two cases are distinguished for deriving this parameter: 1) if the neuron did not spike during the previous Block $n$, this parameter is set to its last value, $a_{i,n}[T_R]$, in the prior Block; otherwise 2), the effect of the spike needs to be taken into account, for which the initial adaptive parameter is expressed as $p_i^m(a_s + p_i^{-1})$. Here, $m$ is the number of time steps remaining in Block $n$ following the spike and $a_s$ is the adaptive parameter value at the time of spiking.

**Proposition 5.** *The adaptive firing threshold $\theta_{i,n+1}[t]$ of neuron $i$ simulated in Block $n+1$ (of length $T_R$) is constructed from the initial adaptive parameter $a_{i,n+1}[0]$, which is equal to its last value in the previous Block if no spike occurred, and otherwise equal to an expression which accounts for the effect of the spike on the adaptive threshold.*

$$\theta_{i,n+1}[t] = 1 + d_i p_i^t a_{i,n+1}[0]$$

$$a_{i,n+1}[0] = \begin{cases} a_{i,n}[T_R] & \text{if } \max_t S_{i,n}[t] = 0 \\ p_i^m(a_s + p_i^{-1}) & \text{otherwise} \end{cases}$$

$$m = \sum_{k}^{T_R} \mathbb{1}_{z_{i,n}[k] > 1}, \quad a_s = \sum_{k}^{T_R} a_{i,n}[k] S_{i,n}[k]$$

---

[3] With appropriate zero padding to the simulation length if it is not divisible by $T_R$.

## 3.3 Theoretical speedup for simulating ALIF SNNs using Blocks

| Method | Computational Complexity | Sequential Operations |
|--------|:------------------------:|:---------------------:|
| Standard | $O(N^{\text{in}} N^{\text{out}} T)$ | $O(T)$ |
| Blocks | $O(N^{\text{in}} N^{\text{out}} T_R^2 N)$ | $O(T/T_R)$ |

Table 1: Computational and sequential complexity of simulating a layer of $N^{\text{out}}$ neurons with $N^{\text{in}}$ input neurons for $T$ time steps using the standard method and our method (with $N$ Blocks each of length $T_R$).

Simulating an ALIF SNN using our Blocks, rather than the conventional approach, requires fewer sequential operations (Table 1). Although the computational complexity of our Blocks approach is larger than the standard approach, the number of sequential operations is less. If we assume that the sequential steps in both methods are executed in an equal amount of time (as all the non-sequential operations can be run in parallel on a GPU), we obtain a theoretical speedup equal to the length of the ARP $\frac{T}{N} = T_R$.

# 4 Experiments

We evaluated the training speedup of our accelerated ALIF SNN relative to the standard ALIF SNN and explored the capacity of our model to be fitted using different spiking classification datasets and real electrophysiological recordings of cortical neurons. Implementation details can be found in the Supplementary material and the code at `https://github.com/webstorms/Blocks`.

## 4.1 Training speedup scales with an increasing ARP

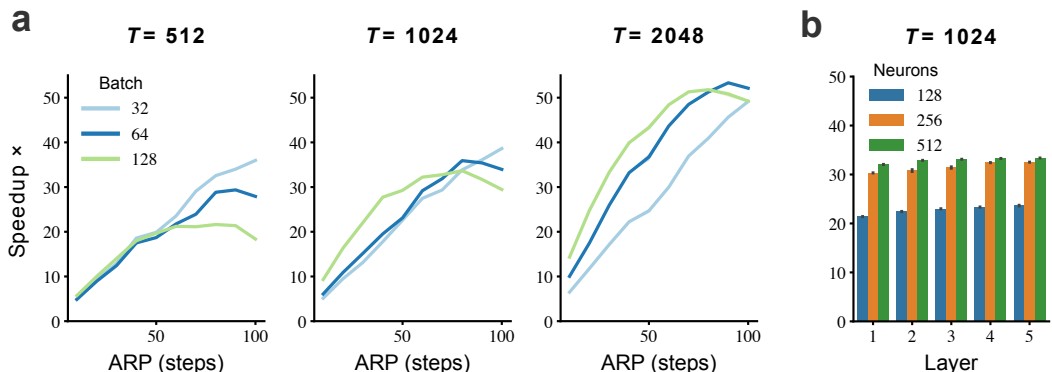

Figure 3: **Training speedup of our model. a.** Training speedup of our accelerated ALIF model compared to the standard ALIF model for different simulation lengths $T$, ARP time steps and batch sizes. **b.** Training speedup over different number of layers and hidden units (with an ARP of $T_R = 40$ time steps, $T = 1024$ time steps and batch size $= 64$; bars plot the mean and standard error over ten runs). Assuming DT= 0.1ms, then 10 time steps = 1ms.

To determine how much faster our model is, we benchmarked the required training duration (forward and backward pass) of our accelerated ALIF SNN to the standard ALIF SNN for varying ARP and simulation lengths using a synthetic spike dataset, with the task of minimizing the number of final-layer output spikes (see Supplementary material). We found the training speedup of our model to increase for a growing ARP (Figure 3a). This speedup was more pronounced for longer simulation durations ($53\times$ speedup for $T = 2048$) than shorter simulation durations ($36\times$ speedup for $T = 512$). These speedups were also present when just running the forward pass (*i.e.* inference using the network; see Supplementary material). Furthermore, we found the speedup of our model to be robust over varying numbers of neurons and layers (Figure 3b). Lastly, we also found our method

to perform the forward pass more than an order of magnitude faster than other publicly available SNN implementations [63, 64] (see Supplementary material).

## 4.2   Accelerated training on spiking classification tasks

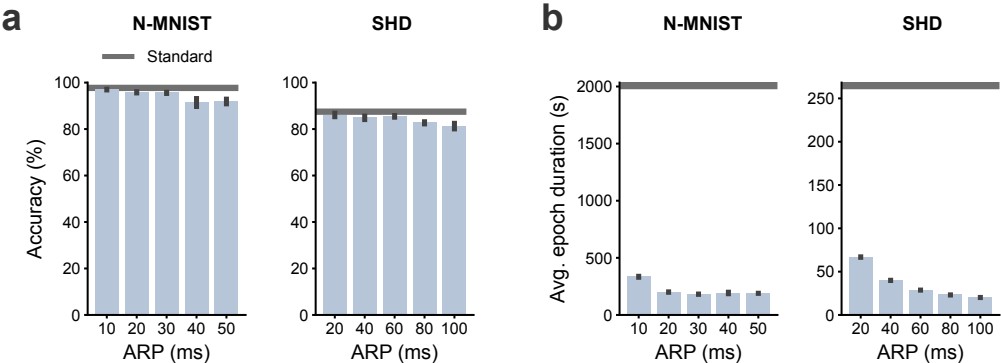

Figure 4: **Performance of our model on spiking datasets. a.** Classification accuracy of our model and the standard ALIF SNN on the N-MNIST and SHD datasets over different (non-biological) ARPs (N-MNIST 1ms=1 time step; SHD 2ms=1 time step). **b.** Training durations of our model and the standard ALIF SNN on the N-MNIST and SHD datasets. Horizontal gray lines plot the standard model's performance (using no ARP) and bars plot the mean and standard error over three runs.

To establish whether our model can learn using backprop with surrogate gradients, and perform on a par with the standard model, we trained our accelerated ALIF SNN and the standard ALIF SNN on the Neuromophic-MNIST (N-MNIST) (using DT= 1ms) [65] and Spiking Heidelberg Digits (SHD) (using DT= 2ms) [66] spiking classification datasets (both commonly used for benchmarking SNN performance [40, 67, 29, 42, 50]). The task of the N-MNIST dataset is to classify spike representations of handwritten digits, and the task of the SHD dataset is to classify spike representations of spoken digits. In all experiments we employed identical model architectures consisting of two hidden layers (of 256 neurons each) and an additional integrator readout layer, with predictions taken from the readout neurons with maximal summated membrane potential over time (as commonly done [66, 68, 29, 42, 50]; see Supplementary material). We adopted the multi-Gaussian surrogate-gradient function [29] to overcome the non-differentiable discontinuity of the spike function (although we found that other surrogate-gradient functions also work, see Supplementary material). Furthermore (as suggested by [68] for conventional SNNs), we only permitted surrogate gradients to flow through the non-recurrent connectivity (which significantly improved performance; see Supplementary material).

We trained our model with different non-biological ARPs on each dataset and contrasted the accuracy and training duration to the standard model (with no ARP). We found a favourable trade-off between classification accuracy and training time for our model. Compared to the standard ALIF SNN model, our model achieved a very similar, albeit slightly lower, accuracy on the N-MNIST (96.97% for ARP=10ms vs standard 97.71%) and SHD dataset (86.07% for ARP=20ms vs standard 87.48%), with the accuracy declining only for the largest ARPs tested (Figure 4a). However, our model only required a fraction of the training time, with an average training epoch of 181s and 20s for the N-MNIST and SHD datasets, respectively, when using the largest tested ARP, compared to the respective standard model training times of 2034s and 268s (Figure 4b).

## 4.3   Quickly fitting real neural recordings on sub-millisecond timescales

We explored the ability of our model to fit *in vitro* electrophysiological recordings from 146 inhibitory and excitatory neurons in mouse primary visual cortex (V1) (provided by the Allen Institute [69, 70]). In these recordings, a variable input current was repeatedly injected at various amplitudes into each real neuron, and the resulting alteration in the membrane potential was recorded (Figure 5a). For each neuron, we used half of the recordings for fitting and the other half for testing, and report all the qualitative and quantitative results on the held-out test dataset. Fitting was achieved by minimising

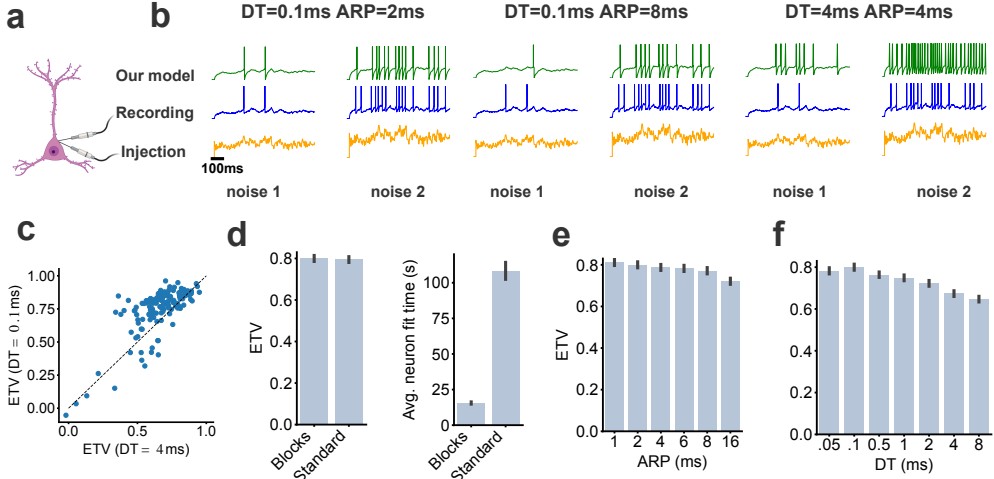

Figure 5: **Fitting cortical electrophysiological recordings. a.** An illustration of a cortical neuron in mouse V1 being recorded whilst stimulated with a noisy current injection. **b.** For held-out data not used for fitting, an example current injection (bottom) and recorded membrane potential (middle) with corresponding fitted model predictions (top). **c.** Comparison of neuron fit accuracy of our model for DT= 0.1ms (y-axis) against DT= 4ms (x-axis). Explained temporal variance (ETV) measures the goodness-of-fit (one is a perfect fit and zero is a chance-level fit). **d.** Comparison of the fit accuracy (left) and duration (right) of our model and the standard model (both using DT= 0.1ms and ARP= 2ms). **e.** Our model's fit accuracy for increasing ARP (with DT= 0.1ms). **f.** Our model's fit accuracy for increasing DT (with ARP=max(2, DT)ms). **d.** to **f.** plots the median and standard error over neurons, except that **d.** (right) plots the mean fit time.

the van Rossum distance between the model and real spike trains [71]. To quantitatively compare the fits of our model using different DTs and ARPs, we employed the explained temporal variance (ETV) measure (used on the Allen Institute website; see Supplementary material). This metric quantifies how well the temporal dynamics of the neurons are captured by the model and takes into account trial-to-trial variability due to intrinsic neural noise. It has a value of one for a perfect fit and a value of zero if the model predicts at chance.

We found that our accelerated ALIF model captured the spike timing of the neurons. Pertinent to the speed-accuracy trade-off, this fit strongly depended on the chosen DT, and less so on the chosen ARP. Qualitatively, using a DT= 0.1ms and an ARP= 2ms, we found our model captured the spike timings of the neurons for current injections of varying amplitude. This still seemed to be the case when we used a larger ARP= 8ms, but the model was worse at capturing spike timings when we used a larger DT= 4ms (with ARP= 4ms; Figure 5b; see Supplementary material for zoomed-in neural traces).

Quantitatively comparing the neuron fits one by one, we found that nearly all of the neuron fits were better when using a DT= 0.1ms (ETV= 0.80) than a DT= 4ms (ETV= 0.66; Figure 5c; using an ARP= 4ms). We examined how accurate and fast our fits were compared to the standard model using an ARP= 2ms and a DT= 0.1ms. Both models achieved a similar ETV of $\sim 0.8$, yet our model only required 15.5s on average per neuron fit time compared to the 108.4s of the standard model (Figure 5d). We further investigated whether a larger ARP could still reasonably fit the data for DT= 0.1ms (to benefit from a faster fit). Consistent with our qualitative observations, we found a less marked reduction in fit accuracy when using larger ARPs (Figure 5e) compared to the drop in performance when using larger DTs (Figure 5f; using an ARP of max(2, DT)ms).

## 5 Discussion

ALIF neurons are a popular model for studying the brain *in silico*. Training these neurons is, however, slow due to their sequential nature [41, 72]. We overcome this by algorithmically reinterpreting the ALIF model. We found that our method permits faster simulation of SNNs, which will likely

play an important role in future research into neural dynamics through simulation [17, 16]. We also confirmed the validity of this approach for modelling neural data. Firstly - of interest to computational neuroscientists - we fitted a multilayered network of neurons using two common spike classification datasets. We found that our model achieved a similar classification accuracy to that of the standard model, even if we increased the ARP to large non-physiological values, which drastically reduced the training duration required for both datasets. Secondly - of interest to computational and experimental neuroscientists - we explored the applicability of our method to fit real electrophysiological recordings of neurons in mouse V1 using sub-millisecond DTs.

We found that our method accurately captured the spike timing of real neurons, fitting their activity patterns in a fraction of the time required by the standard model (although we note that other, more recent spiking-models, might improve fit accuracies [73]). This is particularly important as datasets become larger with advances in recording techniques, requiring faster fitting solutions to characterise computational properties of neurons. As an example of the potential insights provided by our model, we found the fitted V1 neurons to have a heterogenous membrane time constant distribution, suggesting that V1 processes visual information on multiple timescales [42] (see Supplementary material).

Our work will likely also be of interest to neuromorphic engineers and researchers, developing energy-efficient hardware to emulate SNNs [20]. These systems - like real neurons - run in continuous time and thus require training on extremely fine DTs off-chip [74, 75]. Our method could help to accelerate off-chip training times. Furthermore, the ARP hyperparameter in our model can limit high spike rates and thus reduce energy consumption on neuromorphic systems, as this consumption scales approximately proportionally with the number of emitted spikes [76].

A limitation of our method is that the training speedup scales sublinearly - as opposed to linearly - with an increasing ARP simulation length (see Section 3 and 4.1). This is likely due to GPU overheads and employed cudnn routines [77], which further improvements to our code implementation could overcome. An additional limitation is the requirement to define the ARP hyperparameter, whose value influences the training speed and test accuracy in our method and relates to biological realism. However, we found a beneficial trade-off - by using large non-biological ARPs on the artificial spiking classification dataset and small physiological ARPs for the neural fits (although we found larger values to also perform reasonably well) the model achieved comparable accuracy to the standard ALIF model in both cases, while also having greatly increased speed of training and simulation. In particular, the capacity of our model to accurately and quickly fit the data from real neurons is crucial in terms of the biological applicability of this approach.

## Acknowledgments and Disclosure of Funding

We thank Rob Pratt and anonymous reviewers for helpful discussions; and Lorenzo Mazzaschi for feedback on the manuscript. Luke Taylor was supported by the Clarendon Fund. Andrew King and Nicol Harper were supported by the Wellcome Trust (WT108369/Z/2015/Z). Figure 5 was created with BioRender.com.

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
