# Supplementary

## 1 Theoretical proofs

Proofs for all the propositions in the paper, outlining the mathematical equivalence of our Block model to the standard ALIF SNN.

**Proposition 1.** *Membrane potentials without spike reset are computed as a convolution $\tilde{V}_i[t] = (I_i * \tilde{\beta}_i)[t]$ between input current $I_i[t]$ and kernel $\tilde{\beta}_i[t] = (1 - \beta_i)\beta_i^t$ with the initial membrane potential encoded as $I_i[0] = \frac{V_i[0]}{1-\beta_i}$.*

*Proof.* We proceed our proof in two steps. In step 1, we unroll the discretized LIF difference equation (without reset) in time and in step 2, we show how this is equivalent to the proposed convolution. * Step 1, we prove the equivalence between the following equations

$$\tilde{V}_i[t] = \beta_i \tilde{V}_i[t-1] + (1 - \beta_i)I_i[t] \tag{1}$$

$$\tilde{V}_i[t] = \beta_i^t \tilde{V}_i[0] + (1 - \beta_i)\sum_{j=1}^{t} \beta_i^{t-j} I_i[j] \tag{2}$$

We proceed by induction. For $t = 1$ in Equation 2 we obtain

$$\begin{aligned}
\tilde{V}_i[1] &= \beta_i^1 \tilde{V}_i[0] + (1 - \beta_i)\sum_{j=1}^{1} \beta_i^{1-j} I_i[j] \\
&= \beta_i^1 \tilde{V}_i[0] + (1 - \beta_i)I_i[1]
\end{aligned} \tag{3}$$

Hence the relation holds true for the base case $t = 1$. Assume the relation holds true for $t = k \geq 1$, then for $t = k + 1$ we derive

$$\begin{aligned}
\tilde{V}_i[k+1] &= \beta_i \tilde{V}_i[k] + (1 - \beta_i)I_i[k+1] \\
&= \beta_i \Big( \beta_i^k \tilde{V}_i[0] + (1 - \beta_i)\sum_{j=1}^{k} \beta_i^{k-j} I_i[j] \Big) + (1 - \beta_i)I_i[k+1] \\
&= \beta_i^{k+1} \tilde{V}_i[0] + (1 - \beta_i)\sum_{j=1}^{k} \beta_i^{(k+1)-j} I_i[j] + (1 - \beta_i)I_i[k+1] \\
&= \beta_i^{k+1} \tilde{V}_i[0] + (1 - \beta_i)\sum_{j=1}^{k+1} \beta_i^{(k+1)-j} I_i[j]
\end{aligned} \tag{4}$$

This implies equivalence between Equations 1 and 2 for $t = k + 1$ assuming equivalence between Equations 1 and 2 holds true for $t = k$. By the principle of induction, equivalence is established given that both the base case and inductive step hold true.

37th Conference on Neural Information Processing Systems (NeurIPS 2023).

Step 2, as per the proposition, we have

$$\tilde{V}_i[t] = \left(I_i * \tilde{\beta}_i\right)[t]$$

$$= \sum_{j=0}^{t} \tilde{\beta}_i[t-j] I_i[j]$$

$$= (1-\beta_i)\sum_{j=0}^{t} \beta_i^{t-j} I_i[j] \tag{5}$$

$$= (1-\beta_i)\beta_i^t I_i[0] + (1-\beta_i)\sum_{j=1}^{t} \beta_i^{t-j} I_i[j]$$

$$= \beta_i^t \tilde{V}_i[0] + (1-\beta_i)\sum_{j=1}^{t} \beta_i^{t-j} I_i[j]$$

This is identical to Equation 2 and (by step 1) identical to Equation 1. Thus, the proposed convolution in the Proposition computes the membrane potentials without reset. □

**Proposition 2.** *Function $\phi(\tilde{S}_i)[t] = \sum_{k=1}^{t} \tilde{S}_i[k](t-k+1)$ acting on $\tilde{S}_i \in \{0,1\}^T$ contains at most one element equal to one $\phi(\tilde{S}_i)[t] = 1$ for smallest $t$ satisfying $\tilde{S}_i[t] = 1$ (if such $t$ exists).*

*Proof.* Firstly, if $\tilde{S}_i^{(l)}[t] = 0$ for all $t \in [1,T]$ then $\phi(\tilde{S}_i^{(l)})[t] = 0$ for all $t \in [1,T]$ (follows from substitution). Secondly, if $\tilde{S}_i^{(l)}[t_1] = 1$ for smallest $t_1 \in [1,T]$ then $\phi(\tilde{S}_i^{(l)})[t_1] = 1$ (follows from substitution) and there can exist no $t_2 > t_1$ such that $\phi(\tilde{S}_i^{(l)})[t_2] = 1$ as

$$\phi(\tilde{S}_i^{(l)})[t+1] = \sum_{k=1}^{t+1} \tilde{S}_i^{(l)}[k]\big((t+1)-k+1\big)$$

$$= \sum_{k=1}^{t} \tilde{S}_i^{(l)}[k]\big((t+1)-k+1\big) + \tilde{S}_i^{(l)}[t+1]$$

$$= \sum_{k=1}^{t} \tilde{S}_i^{(l)}[k](t-k+1) + \sum_{k=1}^{t} \tilde{S}_i^{(l)}[k] + \tilde{S}_i^{(l)}[t+1] \tag{6}$$

$$= \phi(\tilde{S}_i^{(l)})[t] + \sum_{k=1}^{t+1} \tilde{S}_i^{(l)}[k]$$

Thus $\phi(\tilde{S}_i^{(l)})[t_2] > \phi(\tilde{S}_i^{(l)})[t_1]$ for all $t_2 > t_1$ as $\sum_{k=1}^{t_2} \tilde{S}_i^{(l)}[k] \geq \sum_{k=1}^{t_1} \tilde{S}_i^{(l)}[k] = 1 > 0$. □

**Proposition 3.** *The input current $I_{i,n+1}[t]$ of neuron $i$ simulated in Block $n+1$ (of length $T_R$) is defined as follows, and enforces an absolute refractory period of length $T_R$ and a monosynaptic transmission latency of $D = T_R$.*

$$I_{i,n+1}[t] = \Big( \underbrace{b_i + \sum_{j=1}^{N^{in}} W_{ij} S_{j,n+1}[t]}_{\text{Feedforward current}} + \underbrace{\sum_{j=1}^{N^{out}} W_{ij}^{rec} S_{j,n}[t]}_{\text{Recurrent current}} \Big) \underbrace{\mathbb{1}_{z_{i,n}[t] \geq \max_t S_{i,n}[t]}}_{\text{ARP mask}}$$

*Proof.* We proceed by showing that 1. the transmission latency is the same as in the standard model and 2. the ARP mask enforces an ARP of identical length to the standard model.

1. Identical transmission latency: The relation between the time step $1 \leq t \leq T$ and the time step $1 \leq t_b \leq T_R$ in Block $n \geq 1$ can be expressed as:

$$t = (n-1)T_R + t_b \tag{7}$$

As we assumed the transmission latency $D = T_R$ to be equal to the ARP length, we have

$$t - T_R = ((n-1)-1)T_R + t_b \tag{8}$$

and hence the input current $I_{i,n}[t]$ to Block $n$ at time $t$ (*i.e.* time step $t_b$ in the Block) is computed from the output spikes from the prior Block $n-1$ at the same Block time step $t_b$.

2. Identical ARP length: Case one, if neuron $i$ emitted no spikes during Block $n$, then neuron $i$ should receive input current at every time step in Block $n+1$, which the ARP mask permits (as $z_{i,n}[t] \geq \max_t S_{i,n}[t] = 0$ for all $t$). Case two, if neuron $i$ spiked during Block $n$, then the ARP mask should appropriately mask out the input current to Block $n+1$ (to enforce an ARP of length $T_R$). The number of elements in $z_{i,n}$ which are smaller than one and equal to or larger than one is equal to $T_R$ (can be shown by proof by contradiction):

$$\underbrace{\sum \mathbb{1}_{z_{i,n}[t] < 1}}_{\text{Number of elements masked in Block } n+1} + \underbrace{\sum \mathbb{1}_{z_{i,n}[t] \geq 1}}_{\text{Number of time steps from the spike onwards in Block } n} = T_R \qquad (9)$$

The construction of the Block ensures that at most one spike is emitted within a Block (where $z_{i,n}[t] = 1$) and no further spikes are emitted thereafter (where $z_{i,n}[t] > 1$). The ARP mask ensures that no spikes are emitted in the next Block for the time steps where $z_{i,n}[t] < 1$ (as it only permits input current to flow into the Block for time steps where $z_{i,n}[t] \geq 1$). Thus, the mask ensures enforces the correct ARP length of $T_R$ steps. $\qquad \square$

**Proposition 4.** *The initial membrane potential $V_{i,n+1}[0]$ of neuron $i$ simulated in Block $n+1$ (of length $T_R$) is equal to the last membrane potential in Block $n$ if no spike occurred and zero otherwise.*

$$V_{i,n+1}[0] = \begin{cases} V_{i,n}[T_R], & \text{if } \max_t S_{i,n}[t] = 0 \\ 0, & \text{otherwise} \end{cases}$$

*Proof.* Two cases are distinguished for correctly evolving the membrane potential of a neuron over time. Case one, if no spike occurred in neuron $i$ in Block $n$ (*i.e.* $\max_t S_{i,n}[t] = 0$), then the initial membrane potential of neuron $i$ in Block $n+1$ is equal to the final membrane potential value of neuron $i$ in Block $n$. Otherwise, case two, the initial membrane potential is set to zero (as no state needs to be transferred as the neuron is in an absolute refractory state). $\qquad \square$

**Proposition 5.** *The adaptive firing threshold $\theta_{i,n+1}[t]$ of neuron $i$ simulated in Block $n+1$ (of length $T_R$) is constructed from the initial adaptive parameter $a_{i,n+1}[0]$, which is equal to its last value in the previous Block if no spike occurred, and otherwise equal to an expression which accounts for the effect of the spike on the adaptive threshold.*

$$\theta_{i,n+1}[t] = 1 + d_i p_i^t a_{i,n+1}[0]$$
$$a_{i,n+1}[0] = \begin{cases} a_{i,n}[T_R], & \text{if } \max_t S_{i,n}[t] = 0 \\ p_i^m(a_s + p_i^{-1}) & \text{otherwise} \end{cases}$$
$$m = \sum_k^{T_R} \mathbb{1}_{z_{i,n}[k] > 1}, \quad a_s = \sum_k^{T_R} a_{i,n}[k] S_{i,n}[k]$$

*Proof.* We proceed our proof in two steps. In step 1, we show how the dynamic firing threshold of neuron $i$ in Block $n+1$ can be computed using initial adaptive parameter $a_{i,n+1}[0]$; and in step 2, we show how this initial adaptive parameter is derived.

Step 1, we prove the equivalence between the Block adaptive threshold (Equation 10) and the standard adaptive firing threshold (Equations 11 and 12).

$$\theta_{i,n+1}[t] = 1 + d_i p_i^t a_{i,n+1}[0] \qquad (10)$$
$$\theta_{i,n+1}[t] = 1 + d_i a_{i,n+1}[t] \qquad (11)$$
$$a_{i,n+1}[t] = p_i a_{i,n+1}[t-1] + S_{i,n+1}[t-1] \qquad (12)$$

The spike term in Equation 12 can be dropped, as only the spike occurrence in Block $n$ (and not Block $n+1$) can affect the firing threshold in Block $n+1$ (due to the single spike constraint). Thus,

we rewrite Equation 11 as:

$$\begin{aligned}
\theta_{i,n+1}[t] &= 1 + d_i a_{i,n+1}[t] \\
&= 1 + d_i p_i a_{i,n+1}[t-1] \\
&= 1 + d_i p_i^2 a_{i,n+1}[t-2] \\
&\cdots \\
&= 1 + d_i p_i^t a_{i,n+1}[0]
\end{aligned}$$

Step 2, to derive the initial adaptive parameter $a_{i,n+1}[0]$, we distinguish two cases. Case one, neuron $i$ did not spike in Block $n$, in which case the initial adaptive parameter is set to the final adaptive parameter $a_{i,n}[T_R]$ in Block $n$. Case two, neuron $i$ did spike in Block $n$, and we need to account for the affect of this on the firing threshold in Block $n+1$. If the spike occurred at Block time step $1 \leq T_R - m \leq T_R$ for $0 \leq m < T_R$ we have

$$\begin{aligned}
a_{i,n}[T_R] &= p_i a_{i,n}[T_R - 1] \\
&= p_i^2 a_{i,n}[T_R - 2] \\
&\cdots \\
&= p_i^{m-1} a_{i,n}[T_R - (m-1)] \\
&= p_i^{m-1}\Big( p_i a_{i,n}[T_R - m] + 1 \Big) \\
&= p_i^m \Big( a_{i,n}[T_R - m] + p_i^{-1} \Big) \\
&= p_i^m \Big( a_s + p_i^{-1} \Big)
\end{aligned} \tag{13}$$

with $a_s = \sum_k^{T_R} a_{i,n}[k] S_{i,n}[k] = a_{i,n}[T_R - m]$ (as $S_{i,n}[k]$ is one for $k = T_R - m$ and zero otherwise). Lastly, $m = \sum_k^{T_R} \mathbb{1}_{z_{i,n}[k] > 1}$, as $z_{i,n}[k] > 1$ at ever Block time step $k > T_R - m$ (see Proposition 2 proof showing $z_{i,n}$ to be a strictly increasing sequence if a spike occurred in Block $n$) and thus $\sum_k^{T_R} \mathbb{1}_{z_{i,n}[k] > 1} = \sum_{k=T_R-m+1}^{T_R} 1 = T_R - (T_R - m) = m$. $\qquad\square$

## 2 Experimental details

All models were implemented using PyTorch [1] (although nothing prohibits the use of other auto differentiation frameworks [1, 2]). The speedup benchmarks and neural fits were done on an NVIDIA GeForce RTX 3090, and the training on the spiking classification datasets was done on an NVIDIA GeForce GTX 1080 Ti. Following details apply to both our Block model and standard SNN model which we used as a control.

### 2.1 Speed benchmarks

**Synthetic spike dataset**   We generated binary input spike tensors of shape $B \times N \times T$ ($B$ being the batch size, $N$ the number of input neurons and $T$ the number of time steps). For every batch dimension $b$ a firing rate $r_b \sim \mathbf{U}(u_{\min}, u_{\max})$ was uniformly sampled (with $u_{\min} = 0$Hz and $u_{\max} = 200$Hz – Assuming 1 time step = 1ms), from which a random binary spike matrix of shape $N \times T$ was constructed as a homogenous Poisson process, such that every input neuron in this matrix had a firing rate of $r_b$Hz.

### 2.2 Supervised learning

**Datasets**   We tested our model (and control) on two common spike classification datasets, the Neuromophic-MNIST (N-MNIST) [3] and Spiking Heidelberg Digits (SHD) [4] dataset (both released under the Creative Commons Attribution 4.0 International License). The N-MNIST dataset is the classical MNIST dataset mapped onto a spike code using a neuromorphic vision sensor and the SHD dataset comprises spoken digit waveforms converted into spikes using a model of the auditory bushy neurons in the cochlear nucleus.

**Weight initialisation**  All network connectivity weights were sampled from a uniform distribution $\mathbf{U}(-\sqrt{N^{-1}}, \sqrt{N^{-1}})$ with $N$ number of afferent connections. All biases were initialised as $0$. The hidden neurons were initialised with a membrane time constant of 20ms (*i.e.* $\beta_i^{(l)} = \exp(\frac{-DT}{20})$), an adaptive time constant of 150ms (*i.e.* $p_i^{(l)} = \exp(\frac{-DT}{150})$) and adaptive parameter of $d_i^{(l)} = 1.8$. The readout neurons were initialised with a membrane time constant of 20ms.

**Clamping time constants**  To enforce correct neuron dynamics, we clamped the values of $\beta_i^{(l)}$ into the range $[0.01, 0.99]$ and the values of $p_i^{(l)}$ into the range $[0.0, 0.999]$

$$\beta_i^{(l)} = \begin{cases} 0.99, & \text{if } \beta_i^{(l)} > 0.99 \\ 0.01, & \text{if } \beta_i^{(l)} < 0.01 \end{cases} \tag{14}$$

$$p_i^{(l)} = \begin{cases} 0.999, & \text{if } p_i^{(l)} > 0.999 \\ 0.0, & \text{if } p_i^{(l)} < 0.0 \end{cases} \tag{15}$$

**Readout neurons**  Every network had an output layer of readout neurons (containing the same number of neurons as the number of classes within the dataset trained on), where we removed the spike and reset mechanism (as done in [5]). The output of the readout neuron $c$ in response to input sample $b$ was taken to be the summated membrane potential over time $o_{b,c} = \sum_t V_{b,c}^L[t]$ ($L$ being the readout layer).

**Supervised training loss**  We trained all networks to minimise a cross-entropy loss (with $B$ and $C$ being the number of batch samples and dataset classes respectively)

$$\mathcal{L} = -\frac{1}{B} \sum_{b=1}^{B} \sum_{c=1}^{C} y_{bc} \log(\hat{y}_{bc}) \tag{16}$$

Variable $y_{bc} \in \{0, 1\}^C$ is the one hot target vector and $\hat{y}_{bc}$ are the network prediction probabilities, which were obtained by passing the readout neuron outputs $o_{bc}$ through the softmax function.

$$\hat{y}_{bc} = \frac{\exp(o_{bc})}{\sum_{k=1}^{C} \exp(o_{bk})} \tag{17}$$

**Surrogate gradient**  We tested training on the SHD dataset using three different surrogate gradient functions, including the multi-Gaussian [6], fast sigmoid [7] and the boxcar [8, 9] function - all which have shown to perform well in training SNNs.

$$\frac{\partial S_i^{(l)}[t]}{\partial V_i^{(l)}[t]} = 1.15\mathcal{N}(V_i^{(l)}[t] \mid 0, 0.5^2) - 0.15\mathcal{N}(V_i^{(l)}[t] \mid 3, 3^2) \tag{18}$$

$$- 0.15\mathcal{N}(V_i^{(l)}[t] \mid -3, 3^2) \quad \text{(multi-Gaussian)} \tag{19}$$

$$\frac{\partial S_i^{(l)}[t]}{\partial V_i^{(l)}[t]} = (10|V_i^{(l)}[t]| + 1)^{-2} \quad \text{(fast sigmoid)} \tag{20}$$

$$\frac{\partial S_i^{(l)}[t]}{\partial V_i^{(l)}[t]} = \begin{cases} 0.5, & \text{if } |V_i^{(l)}[t] - \theta_i^{(l)}| \leq 0.5 \\ 0, & \text{otherwise} \end{cases} \quad \text{(boxcar)} \tag{21}$$

$$\tag{22}$$

We chose the multi-Gaussian function as this obtained the best performance (Supplementary Figure 1a).

**Detaching gradients**  Detaching gradients from flowing through the spike reset and recurrent connections has been shown to improve classification accuracies when training SNNs [5]. We thus explored 1. allowing gradients to flow through all elements within the computational graph (*i.e.* attached) and 2. detaching the surrogate gradient from all elements within the computational graph, except for the feedforward connections to other neurons (*i.e.* detached). We found that detaching improved test accuracies across all the tried surrogate gradient functions on the SHD dataset (Supplementary Figure 1b).

**Training procedure** We used the Adam optimiser (with default parameters) [10] for all training, starting with an initial learning rate of 0.001, which was decayed by a factor of 10 every time the number of epochs reached a new milestone. Model weights were saved if the training error at the end of each epoch was lowered. The hyperparameters are found in Supplementary Table 1.

**Additional results** Forward (*i.e.* inference using the network) speedup can be found in Supplementary Figure 2. Here we used the same simulation setup as the one we used for benchmarking the training speedups. We also compared the forward pass of our model to other publicly available SNN implementations, the Norse library [11] and the Spiking Jelly library [12], and found our method to run considerably faster (Supplementary Table 2). We benchmarked a simple one layer SNN network of 100 units on the forward pass, over 1000 time steps with 200 input units (with a batch size of 128). As expected, we found the Norse (0.30s) and SpikingJelly (0.18s) implementations took a similar time to our standard SNN implementation (0.33s) with the SpikingJelly implementation running slightly faster (as they are all governed by the same sequential time complexity). In contrast, our Blocks model (using a 50 steps ARP) ran over an order of magnitude faster (0.016s) than the other implementations.

## 2.3 Neural fits

**Dataset preprocessing** We fitted the model to *in vitro* whole-cell patch clamp electrophysiological recordings from neurons in layer 4 in mouse primary visual cortex, provided by the Allen institute [13, 14]. These neurons were injected with a varying current at different amplitudes. For each neuron, the training and test datasets were those stipulated by the Allen Institute (50% for training and 50% for testing). We only fitted and reported our results on neurons which had four repeats for each unique current injection. We processed the input data by 1. removing all the long periods in the recordings during which no current was injected, 2. resampling the data to have a DT$= 0.1$ms, and lastly 3. normalising the data by subtracting the mean and by dividing by the standard deviation of the training dataset. We used the spike times provided by the Allen Institute for fitting.

**Weight initialisation** The neuron input weight was initialised to a constant value of $\frac{s}{100}$ for scale value $s$, ensuring that the input weight was correctly rescaled for different simulation resolutions (*e.g.* $s = 1$ corresponds to simulating with DT$= 0.1$ms and $s = 2$ corresponds to simulating with DT$= 0.2$ms). The bias was initialized to 0. The membrane time constant was initialized to 20ms, the adaptive time constant was initialized to 100ms, and the adaptive parameter was initialized to $d = \frac{0.1}{s}$.

**Supervised training loss** Each ALIF neuron model was optimised to produce the same spike times of the real neuron being fit to. This was achieved by minimising the van Rossum distance $D_R$ [15] between the predicted model spike train $x$ and real neuron spike train $y$, defined as

$$D_R(\tau_R) = \sqrt{\frac{1}{\tau_R} \int_0^T \left(k * x(t) - k * y(t)\right)^2 dt} \tag{23}$$

with exponential kernel $k = H(t) \exp(\frac{-t}{\tau_R})$ and $H$ being the heaviside function. We used $\tau_R = 100$ms for all our reported results.

**Training procedure** We used the Adam optimiser (with default parameters) [10] for every neuron fitted, with a learning rate of 0.0001. Training was carried out over 200 epochs in full batch mode (*i.e.* estimating gradients from the entire training dataset). Model parameters were saved whenever the training score improved, and training was halted if there was no improvement over the last five epochs.

**Explained temporal variance** We used the explained temporal variance (ETV) metric to assess the goodness-of-fit of the fitted models to the neural data (used on the Allen Institute website). The metric has a value of zero when the model predicts at chance and a value of one for a perfect fit to the data, and is defined as:

$$\text{ETV} = \frac{\text{ETV}_{\text{raw}}}{\text{ETV}_{\text{max}}} \tag{24}$$

ETV$_{\text{raw}}$ measures the pairwise explained variance of the data with the model and ETV$_{\text{max}}$ measures the upper limit on how well the model can perform (*i.e.* how much of the neuron's response variability from repeat to repeat can be accounted for).

$$\text{ETV}_{\text{raw}} = \sum_r \frac{\text{var}\,(g * x) + \text{var}\,(g * y_r) - \text{var}\,(g * x - g * y_r)}{\text{var}\,(g * x) + \text{var}\,(g * y_r)} \tag{25}$$

$$\text{ETV}_{\text{max}} = \sum_r \frac{\text{var}\,(g * \bar{y}) + \text{var}\,(g * y_r) - \text{var}\,(g * \bar{y} - g * y_r)}{\text{var}\,(g * \bar{y}) + \text{var}\,(g * y_r)} \tag{26}$$

Here $x$ is the predicted model spike train, $y_r$ is recorded neuron spike train for repeat $r$, $g$ is a Gaussian kernel (with mean $\mu = 0$ and standard deviation $\sigma = 150$ms) and $\bar{y}[t] = \frac{1}{R} \sum_r^R (g * y_r)[t]$ is the mean recorded (and smoothed) neuron response. Convolving the spike trains with the Gaussian kernel converts them into peristimulus time histograms.

**Additional results**   Zoomed-in plots of the neural traces can be found in Supplementary Figure 3, and the membrane time constant distribution of the fitted V1 neurons can be found in Supplementary Figure 4.

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

Table 1: **Dataset and corresponding training parameters.**

|  | N-MNIST | SHD |
|---|---|---|
| Dataset (train/test) | 60k/10k | 8156/2264 |
| Input neurons | 1156 | 700 |
| Dataset classes | 10 | 20 |
| Epochs | 30 | 40 |
| Batch size $B$ | 64 | 64 |
| Time steps $T$ | 300 | 600 |
| Time resolution $\Delta t$ (ms) | 1 | 2 |
| LR decay epoch milestones | $N/A$ | $(15, 30)$ |

Table 2: **Comparison to other SNN libraries**. Forward pass simulation duration of our model compared to other publicly available SNN implementations.

|  | Duration (s) |
|---|---|
| SpikingJelly [11] | 0.18s |
| Norse [12] | 0.30s |
| Standard SNN (our implementation) | 0.33s |
| Block SNN (our model) | 0.016s |

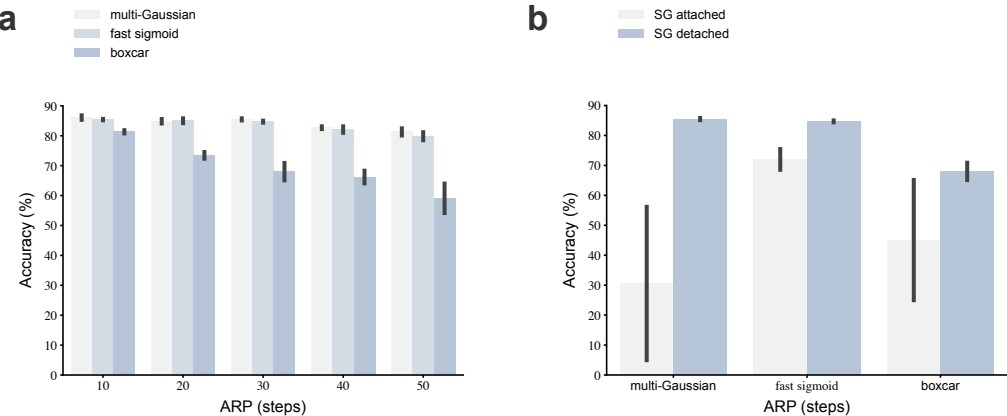

Figure 1: **Surrogate gradient search. a.** Accuracy on the SHD dataset using our model for different surrogate gradient functions. **b.** Accuracy on the SHD dataset using our model when attaching the surrogate gradient to all elements within the computational graph vs detaching it from everything besides the connections to efferent neurons. Bars plot the mean and standard error over three runs.

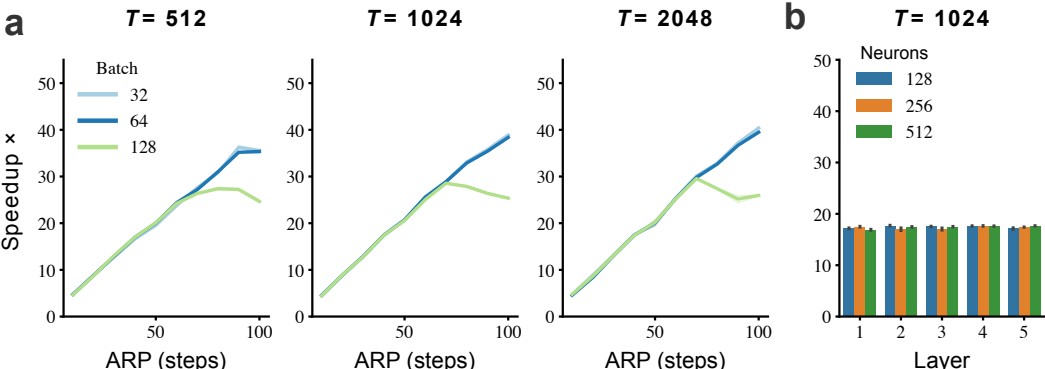

Figure 2: **Forward speedup of our model. a.** Forward speedup of our accelerated ALIF model compared to the standard ALIF model for different simulation lengths $T$, ARP time steps and batch sizes. **b.** Forward speedup over different number of layers and hidden units (with ARP time steps= 40, $T = 1024$ and batch= 64; bars plot the mean and standard error over ten runs). Assuming DT= 0.1ms, then 10 time steps = 1ms.

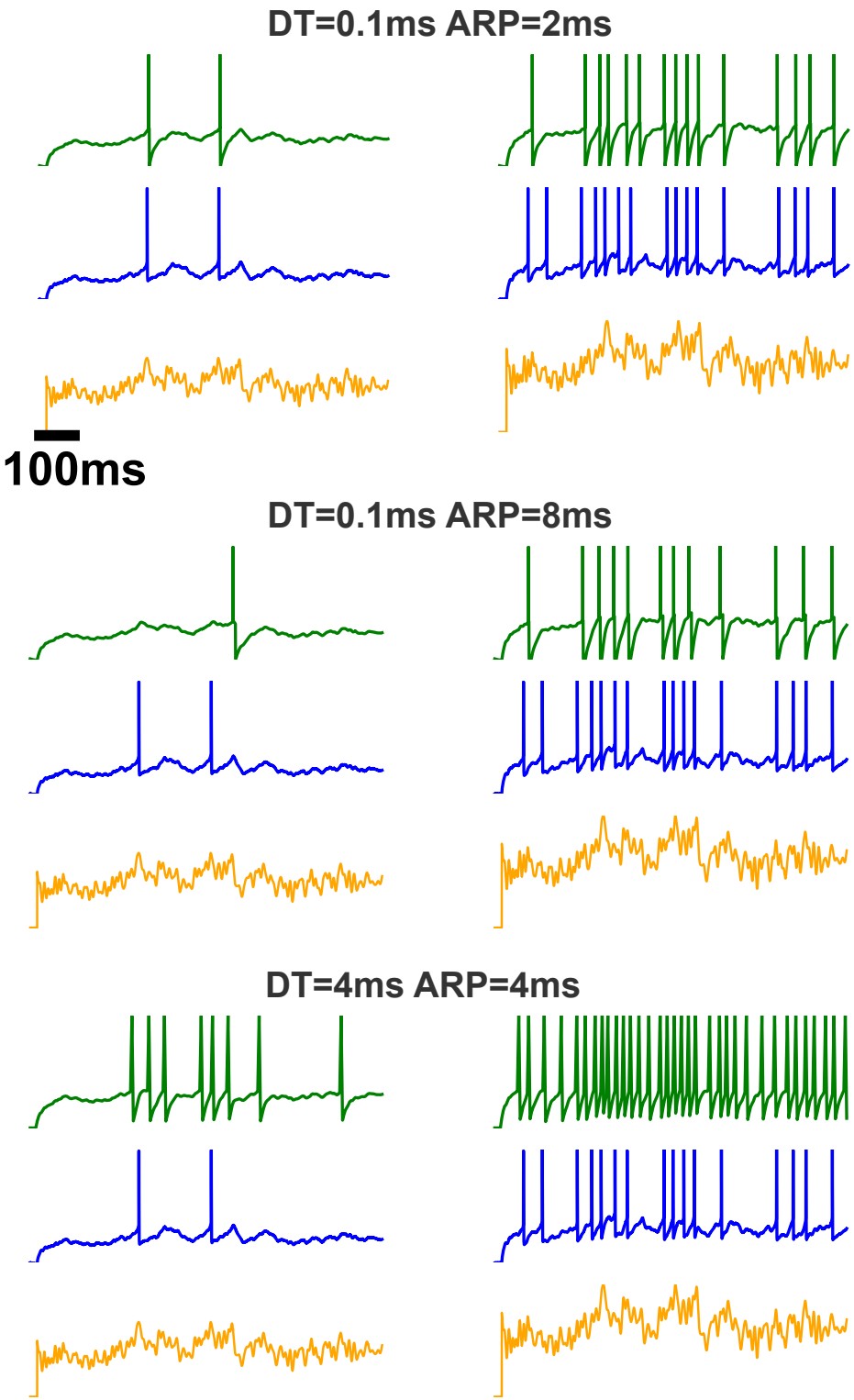

Figure 3: **Zoomed-in versions of the neural traces from Figure 5b.**

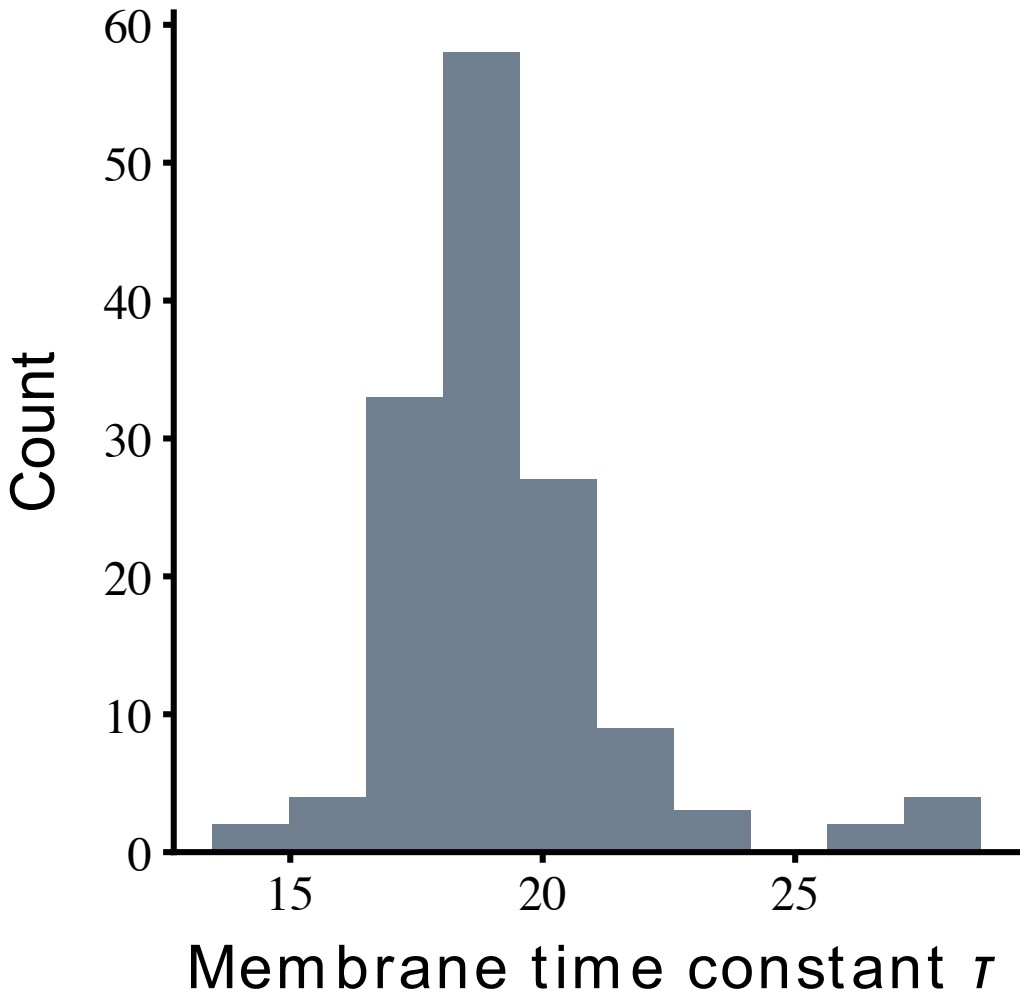

Figure 4: **Membrane time constant distribution of the neurons fitted in Figure 5.**