# OpenReview forum: "Addressing the speed-accuracy simulation trade-off for adaptive spiking neurons"
_NeurIPS.cc/2023/Conference — NeurIPS 2023 poster_

### Official Review · Reviewer_Fo7F · 2023-07-06

**Soundness:** 2 fair
**Presentation:** 2 fair
**Contribution:** 2 fair
**Rating:** 5
**Confidence:** 3

**Summary:**

This paper focus on addressing the speed-accuracy tradeoff in adaptive spiking neurons. They utilize the existence of the absolute refractory period (ARP), and the fact that a neuron can spike at most once within this period, which could reduce the complexity to O(1), and develop an algorithmic reformulation of the adaptive ALIF model. Their model is simulated in blocks of time, instead of simulating network dynamics step by step. Their model achieves ~40x speed up in inference and ~53x speed up in training without the sacrificing the accuracy. And the model is also tested on real electrophysiological recording.

**Strengths:**

1. Motivation:
The paper is well-motivated, and addressing the speed-accuracy tradeoff in the biological simulation is an important question. And using the fact that neuron can spike at most once within the absolute refractory period (ARP) to speed up the computation is novel.
2. Evaluation:
Two levels of evaluations were properly designed, one is to train SNN on spiking classification tasks (N-MNIST, SHD datasets), and the other one is to fit real neural recordings.
3. Result:
Their method is mathematically proved that significantly improve the speed without sacrificing the accuracy, and it's also evaluated by multiple experiments at the same time.

**Weaknesses:**

1. Generalization: One major weakness is that the proposed algorithm using the sparse firing in ARP to speed up the computation is quite limited to one particular type pf SNN, and hard to be generalized to other variants of neural network simulation.
2. Method: The method's performance depends on the value of hyper parameters ARP and DT, and their values are manually tuned for different experiments, which is not practical for real applications, where the ARP is typically unknown and varied across neurons. The method will be more efficient if these hyperparameters could be automatically optimized and learned.
2. Evaluation: No additional baseline was evaluated and compared under the same experiments.

**Questions:**

1. How does this method could be generalized to standard spiking neurons models (leaky integrate-and-fire model without adaptive firing threshold, Hodgkin–Huxley model, etc.)?
2. How does the speed compared to continuous neural dynamical models such as neural ODE?
3. What's the additional overhead time and memory cost to use this algorithm?


**Limitations:**

1. No potential societal impact of the work.
2. The ARP hyperparameters is not addressed, an automatically tuning algorithm might be potentially helpful.

---

> ### Author Rebuttal · Authors · 2023-08-08
>
> > Motivation: The paper is well-motivated, and addressing the speed-accuracy tradeoff in the biological simulation is an important question. And using the fact that neuron can spike at most once within the absolute refractory period (ARP) to speed up the computation is novel.
>
> Thank you for acknowledging the importance of our work and the novelty of our solution.
>
> > Generalization: One major weakness is that the proposed algorithm using the sparse firing in ARP to speed up the computation is quite limited to one particular type pf SNN, and hard to be generalized to other variants of neural network simulation.
>
> We respectfully disagree. Extending our work to other variants of neural network simulation is indeed an exciting avenue for future research, but this goes beyond the scope of the present study. We have managed to accelerate one of the most commonly used spiking models: the ALIF (and as a consequence the LIF) model. These models have been instrumental to computational neuroscientists and neuromorphic engineers, and will thus be of great interest to many researchers. We also show that the sparseness - as a result of the ARP - is not necessarily a limiting factor, with our method  achieving good performance on both the ML benchmarks (original Figure 4) and the neural fits (original Figure 5). We think this represents a significant advance (as noted in the other reviews).
>
> > Method: The method's performance depends on the value of hyper parameters ARP and DT, and their values are manually tuned for different experiments, which is not practical for real applications, where the ARP is typically unknown and varied across neurons. The method will be more efficient if these hyperparameters could be automatically optimized and learned.
>
> The DT hyper-parameter has to be defined irrespective of the method used, and not knowing its optimal values or having an automated optimization method is not a shortcoming of our work alone, but applies more generally to all SNN methods. Researchers currently use known and tried values, like 0.1ms and 1ms DT for neural fits and machine learning benchmarks respectively (Perez-Nieves and Goodman, 2023). The ARP is not commonly used in SNN literature, but it is known to be approximately 1-2ms in real neurons (see line 86 for references). We also experimentally showed that extending the ARP to large values results in negligible degradation in performance accuracy on the ML benchmarks (original Figure 4) and in the neural fits (original Figure 5), thus demonstrating the robustness of our method to this hyper-parameter. We agree that learning the ARP hyper-parameter would be impactful to SNN research as a whole.
>
> > Evaluation: No additional baseline was evaluated and compared under the same experiments.
>
> Could you please be more specific, and we can try to address any additional baselines you have in mind? We have evaluated our method on well-known machine learning benchmarks and have explored its applicability to fitting real neural recordings, which we would argue to be a respectable amount of experimental validation (especially in the context of the novel theoretical contributions of our work).
>
> > How does this method could be generalized to standard spiking neurons models (leaky integrate-and-fire model without adaptive firing threshold, Hodgkin–Huxley model, etc.)?
>
> The ALIF is a generalisation of the LIF and thus all our work also applies to the LIF (just set $d_i^{(l)}=0$ in Equation 5 to see why). The Hodgkin–Huxley (HH) model is more complex than the LIF and ALIF models (with the neuron dynamics defined by a set of ordinary differential equations). Future work might be able to apply aspects of our work to accelerate the HH model; however this is beyond the scope of our submission.
>
> > How does the speed compared to continuous neural dynamical models such as neural ODE?
>
> This comparison is not appropriate, as neural ODE models - as far as we are aware of - are non-spiking, and the premise of our work is to accelerate spiking neural networks.
>
> > What's the additional overhead time and memory cost to use this algorithm?
>
> Our method introduces more computation (requiring more memory), but reduces sequential processing for increasing ARP (requiring less memory). We found our method to actually require less memory on the neural fitting task (using an ARP=2ms and a DT=0.05ms) run on a NVIDIA GeForce RTX 3090  (our method: 1579MB vs standard method: 1699MB). We acknowledge that memory requirements will differ for different GPUs and different block lengths.
>
> > The ARP hyperparameters is not addressed, an automatically tuning algorithm might be potentially helpful.
>
> This concern has been addressed above. As you suggest, a tuning algorithm might potentially be helpful, but it is known that the ARP is approximately 1-2ms in real neurons (see line 86 for references) and we show here that increasing this to non-biological values has little impact on the performance of our method when tested against either ML benchmarks or real neural recordings. We do not think that the absence of an automatic tuning algorithm is a major limitation or detracts from our novel contribution of describing a new accelerated training method, vetted on ML and neural-fitting tasks, which should be of interest to many SNN researchers.

---

> > ### Comment · Reviewer_Fo7F · 2023-08-14
> >
> > Thanks for the reviewers' response. Most of my concerns has been adequately addressed, especially related to overall improvement of speed without significant sacrifice in accuracy, robustness of hyperparameters on accuracy. Future extensions to more recent or other variants of spiking-models would still be valuable. Therefore, I changed my score to borderline accept accordingly.

---

### Official Review · Reviewer_9z9h · 2023-07-07

**Soundness:** 3 good
**Presentation:** 4 excellent
**Contribution:** 2 fair
**Rating:** 5
**Confidence:** 4

**Summary:**

The submitted manuscript proposed a method to speedup simulations of adaptive leaky integrate-and-fire (ALIF) neurons on a GPU by using the fact that during the absolutely refractory period ARP, neurons don't exchange any information and thus "blocks" on dt/ARP can be parallelized on GPU. This approach is then used to both train ALIF neurons on typical machine learning tasks and to fit individual ALIF neuron parameters to experimentally measured membrane potentials/currents/spike trains from the Allen Institute, with demonstrate a speedup achieved by the algorithm.

Overall, the submission is well-written and the numerical experiments seem to be executed with due diligence. Unfortunately, no code is provided, which makes the claims and simulations not reproducible.
The core idea to parallelize the time of the ARP is not new and has been used by other simulators (e.g. the NEST simulator for MPI parallelization as described here: https://nest-simulator.readthedocs.io/en/v2.20.0/guides/running_simulations.html). However, I have not seen it explicitly used for GPU parallelization and training.
While the novelty of the submission seems to be minor, I would still think that it can be a useful contribution to the field once the code is made available and the implementation is benchmarked against existing implementations. Once the code is made available and benchmarks are added, I am willing to improve my score and recommend the paper for publication at NeurIPS.

UPDATE POSTREBUTTAL:
The code is now made available.  While the originality of the contribution is somewhat limited, the preliminary benchmarks suggest that it might be a useful contribtion to the spiking network community.

**Strengths:**

* The manuscript is written clearly.
* The manuscript is relevant for both the 'spiking machine learning' community (with the ML tasks) and the neuroscience community with the neuron-fitting examples.
* The implementation might be helpful for the community, as this trick seems not to be widely used yet.
* figure 4.2 is great

**Weaknesses:**

* The core idea seems not to be new: As described above, the idea to parallelize the time of the ARP was used by other simulators (e.g. the NEST simulator for MPI parallelization as described here: https://nest-simulator.readthedocs.io/en/v2.20.0/guides/running_simulations.html). It would be crucial to cite this previous work and/or describe what is new/different here.
* The NEST parallelization seems to scale close to linear, thus the presented manuscript might even be not state of the art.
* For the fitting of neuron models, ALIF is not state of the art (see for example Pozzorini et al. for a more recent benchmark of fitting different neuron models 2015 https://doi.org/10.1371/journal.pcbi.1004275). Currently, the manuscript claims "ALIF neurons have been shown to accurately fit real neural recordings." in line 32, and cites a paper from 2008 that is not state-of-the-art. Please reference a more recent paper and mention the caveat that the ALIF is not state-of-the art for fitting neuronal dynamics.
* The notation O(T) compared to (T/T_R) slightly confusing. It seems to compare quantities with different units (assuming T and T_R have units of time). Does the speedup not also depend on DT? After reading more carefully, I guess what you actually want to claim is that the computational costs scale proportional to T and that during blocks of size T_R/DT no communication between threads has to occur which can lead to a theoretical maximal speedup of min(N_threads,T_R/DT) where is the number of parallel threads of the GPU or CPU.

Minor issues:
fix latex of N^in

**Questions:**

* How does the speedup depend on the number of parallel threads and on dt?
* Put the value of ARF in the caption of Figure 4a. Was it 100 steps, so 10 ms?? How biologically realistic is an ART of 10ms?
* * Figure 4.3 time axis labels missing! Can you put a zoomed-in version in the supplement with finer time-axis so the artifacts introduced by large ART are easier to see?
* Figure 5c: does ETV stand for? Please write in the caption, it is only defined in the supplement.
* figure 5d: What is ETV for Dt=0.05ms? Maybe you have even better performance (and better speedup according to the reasoning above) for smaller dt? ;)
* Why in 5f is blocks better than standard? shouldn't they be the same?
* 5e: What happens for shorter ARP?

**Limitations:**

* No code is provided, therefore the numerical results are unfortunately not reproducible. Also, the code will make the submission substantially more useful for the community.  If code is provided during the review, I will reconsider my score.
* ALIF is not state-of-the-art (see above).
* using ARP for parallelization speedup is not strictly new (see above)

---

> ### Author Rebuttal · Authors · 2023-08-08
>
> > Once the code is made available, I am willing to improve my score and recommend the paper for publication at NeurIPS.
>
> Thank you! We have uploaded all the code, including instructions to replicate all the experimental findings for the machine learning and neural-fitting benchmarks. In addition, we have included a getting-started notebook to help newcomers with the model.
>
> Addressing your specific concerns and questions:
>
> > The core idea seems not to be new: As described above, the idea to parallelize the time of the ARP was used by other simulators (e.g. the NEST simulator for MPI parallelization [...]). It would be crucial to cite this previous work and/or describe what is new/different here.
>
> Thank you for making us aware of this work. We have now cited the NEST library in the related work section of our manuscript (Section 2). A key apparent difference is that our method supports simulation and training by backprop, whereas the NEST simulator seems to just support simulation but not training by backprop. Their speedup technique also seems bound to CPUs, whereas our method supports faster training on GPUs. Also, while NEST does transmit spikes between units at extended intervals, it does not appear to use a block update method for the membrane potential like we do, but rather to update the state of the neurons at every simulation time step.
>
> > The NEST parallelization seems to scale close to linear, thus the presented manuscript might even be not state of the art.
>
> How scaling of NEST with parallelization (Morrison et al., 2005) relates to the scaling of our model with block length is not immediately apparent. However, our method scales linearly with the block length when simulating (but not training) ALIF SNNs for moderate batch sizes (see Figure 1 in the Supplementary). Regardless, our method is state of the art with respect to substantially speeding up the training of SNNs.
>
> > For the fitting of neuron models, ALIF is not state of the art (see for example Pozzorini et al. for a more recent benchmark of fitting different neuron models 2015 [...]). Currently, the manuscript claims "ALIF neurons have been shown to accurately fit real neural recordings." in line 32, and cites a paper from 2008 that is not state-of-the-art. Please reference a more recent paper and mention the caveat that the ALIF is not state-of-the art for fitting neuronal dynamics.
>
> LIF and ALIF models are very commonly used models of biological neurons. We have included more recent publications showcasing ALIF neuron fits to real neural recordings (in the introduction), and have mentioned the caveat that there are more recent spiking-models available that can be fit to neural responses (in the Discussion), citing Pozzorini et al., 2015. We note that Pozzorini et al., 2015 also does not compare their model with the ALIF model, but with the GLM model.
>
> > The notation O(T) compared to (T/T_R) slightly confusing. It seems to compare quantities with different units (assuming T and T_R have units of time). Does the speedup not also depend on DT?
>
> The speedup does depend on DT (smaller DT leads to faster speedup). T and T_R are in time steps, not in milliseconds. We have edited the manuscript in places where this is ambiguous and have added a footnote on page 1 to make this explicit.
>
> > After reading more carefully, I guess what you actually want to claim is that the computational costs scale proportional to T and that during blocks of size T_R/DT no communication between threads has to occur which can lead to a theoretical maximal speedup of min(N_threads,T_R/DT) where is the number of parallel threads of the GPU or CPU.
>
> This is correct (assuming all the computation of a block can run on a single thread).
>
> > How does the speedup depend on the number of parallel threads and on dt?
>
> Addressed in the last two answers (although we can expand more if you like).
>
> > Put the value of ARF in the caption of Figure 4a. Was it 100 steps, so 10 ms??
>
> We have now defined the correspondence between simulation step and DT in the Figure caption (N-MNIST 1ms=1 sim. step; SHD 2ms=1 sim. step). See new Figure 4.
>
> > How biologically realistic is an ART of 10ms?
>
> Not very, neurons usually have an ARP of 1 - 2ms (see line 86), as we use in Figure 5 for modelling real neurons.
>
> > Figure 4.3 time axis labels missing! Can you put a zoomed-in version in the supplement with finer time-axis so the artifacts introduced by large ART are easier to see?
>
> As requested, we have now added a time axis (see new Figure 5b) and added a zoomed-in version in the supplement.
>
> > Figure 5c: does ETV stand for? Please write in the caption, it is only defined in the supplement.
>
> Done. ETV stands for explained temporal variance (also defined on line 242).
>
> > figure 5d: What is ETV for Dt=0.05ms? Maybe you have even better performance (and better speedup according to the reasoning above) for smaller dt? ;)
>
> We have run this additional experiment. We did not see an improvement in performance (see new Figure 5f). However, as expected, there was a drastic change in speedup between our method and the standard method, from x6.9 (using DT=0.1ms) to x12.9 (using DT=0.05ms).
>
> > Why in 5f is blocks better than standard? shouldn't they be the same?
>
> In response to a request by reviewer #1, this is now Figure 5d. Exchanging learned weights between either model would result in the same performance. However, slight differences can arise during training, due to different random weight initialization, and potentially due to differences in the computational graphs between the methods (resulting in different gradients).
>
> > 5e: What happens for shorter ARP?
>
> We have run the additional experiment and found a shorter ARP to slightly increase performance (see new Figure 5e).
>
> > Minor issues: fix latex of N^in
>
> Fixed (assuming this was in the caption of Table 1).

---

> > ### Comment · Reviewer_9z9h · 2023-08-14
> > **acknowledgement of rebuttal // final question**
> >
> > The detailed responses are appreciated.
> >
> > *    Code Availability: The reviewer has not been able to access the provided code. It is recommended that the area chair be contacted to ensure the code is made available for review.
> >
> > *    Parallelization of ARP: The differentiation from the NEST simulator is appreciated. It is essential that the NEST algorithm is cited appropriately in the manuscript.
> >
> > *    State of the Art: Reservations remain due to the absence of benchmarking against other spiking network simulators and AD implementations. However, the benefit of the doubt is given. A more detailed implementation and benchmarking would enhance the manuscript's strength.
> >
> > *    Neuron Model Fitting: The absence of ALIF in Pozzorini is noted.
> >
> > For points 5 to 11, the clarifications and additional experiments conducted have addressed the concerns.
> >
> > Computational Graph Differences: The rebuttal mentions potential differences in the computational graphs between methods, leading to different gradients. An elaboration on the specific differences in the computational graph would be beneficial.
> >
> > In conclusion, while there are a few remaining reservations, significant efforts have been made to address the majority of the concerns raised.

---

> > > ### Author Response · Authors · 2023-08-15
> > >
> > > Thank you for your response. We have reminded the area chair to share the code with you and the other reviewers. We now cite the NEST framework and key papers (Morrison et al., 2005; 2007) in the related work section of the paper.
> > >
> > > PyTorch - the machine learning framework we used - creates a computational graph for a given model, where every node therein contains one mathematical expression of the model. These graphs are then used to automatically calculate the gradients in a model with respect to a given loss function. Although the standard model (Section 2) and our model (Section 3) implement exactly the same ALIF dynamics, they are composed of different mathematical expressions (contrast standard Equations 1-5 with our block equations 6 onwards), which thus results in different computational graphs and potentially slightly different gradients. However, graph differences have little or no effect on the trained model, as evidenced by the block and the standard models having very similar performance scores (original Figure 5f).
> > >
> > > We thank you again for your readiness to raise your score and to recommend the paper for publication at NeurIPS. We hope our clarifications and further experiments are sufficient to merit a score raise.

---

> > > ### Author Response · Authors · 2023-08-20
> > >
> > > Dear reviewer, with the discussion period coming to an end, we hope we were able to address your remaining reservations, and hopefully with access to our code, you are now willing to recommend the paper for publication. We thank you for your time.

---

> > > > ### Comment · Reviewer_9z9h · 2023-08-21
> > > > **acknowledgement of code // reservations persist**
> > > >
> > > > I acknowledge that the code is now made public. After briefly inspecting the code, my reservations persist:
> > > > * The idea to parallelize the time of the ARP is not new and is already being used for many years. Even if details of the implementation are different, the overall idea is the same. Thus, my reservations regarding originality persist. The fact that the authors now used the established trick also in the backward pass using automatic differentiation is for me not a major conceptual innovation.
> > > > * Benchmarks missing: It still remains unclear if the methods is actually competitive to other implementations both for the forward pass and for training (backward pass). Without comparison to existing implementations (e.g. Norse (https://github.com/norse/norse) or GeNN (https://github.com/genn-team/genn) or SpikingJelly (https://github.com/fangwei123456/spikingjelly), the practical relevance of the suggested method remains unclear. Benchmarking would enhance the manuscript's strength.

---

> > > > > ### Author Response · Authors · 2023-08-21
> > > > >
> > > > > Unfortunately, we were unable to find any technical reports detailing the use of the ARP to speedup SNN simulation and training on GPUs. Our method runs considerably faster than the Norse and SpikingJelly implementations which you have listed (we did not have time to run GeNN as this requires a long list of installation steps). We benchmarked a simple one layer SNN network of 100 units on the forward pass, over 1000 simulation steps with 200 input units (with a batch size of 128), and obtained the following results:
> > > > >
> > > > > Norse: 0.30s
> > > > >
> > > > > SpikingJelly: 0.18s
> > > > >
> > > > > Standard SNN (our implementation): 0.33s
> > > > >
> > > > > Blocks (our model – using 50 steps for ARP): 0.016s
> > > > >
> > > > >
> > > > > As expected, we found the Norse and SpikingJelly to obtain a similar time to our standard SNN implementation, with the SpikingJelly running slightly faster (as they are all governed by the same sequential time complexity), whereas our Blocks model runs over an order of magnitude faster. You can find a notebook of the results, with the code, on the original link posted.

---

> > > > > > ### Comment · Reviewer_9z9h · 2023-08-22
> > > > > > **acknowledgement of code and benchmark**
> > > > > >
> > > > > > Thank you for the detailed explanation, code and benchmarks! Based on the authors' recent experiments and additional clarifications, I'm pleased to adjust my rating!

---

> ### Comment · Area_Chair_JUfv · 2023-08-18
> **Code**
>
> Dear reviewer,
>
> the authors shared the code at an anonymized GitHub repository:
> Link to anonymized Github repo: https://github.com/webstorms/Blocks
>
> Best regards,
>   The area chair

---

### Official Review · Reviewer_x3Zh · 2023-07-12

**Soundness:** 2 fair
**Presentation:** 3 good
**Contribution:** 3 good
**Rating:** 7
**Confidence:** 3

**Summary:**

The authors implement an adaptive leaky integrate-and-fire model in a parallelized way on GPUs by incorporating a fixed refractory period that enables a window of time (‘block’) to be processed in one go. This accelerates the inference and training of spiking neuron models, while retaining accuracy, as shown by the authors.

**Strengths:**

The authors provide an impressive array of results for speedups for training and inference of spiking neural networks (SNNs) and real neural recordings. The accuracy does not seem to decrease although a refractory period is introduced. This may have great implications for future inference of SNNs. The paper on the whole is written clearly.

**Weaknesses:**

It is unclear why the accuracy does not go down in the empirical results presented. If the input signal is not being integrated in the case of the refractory period, then that part of the signal is forever lost. This should naturally lead to a decrease in accuracy by the network. Please elaborate why this is not happening. Is this something specific to the tasks that you are performing, and how is the accuracy impacted in general? Simple simulations with a couple of neurons may help here, while varying the ARP. This may also give some intuition as to how to pick this number to minimize loss of accuracy while still obtaining speedups.

**Questions:**

See ‘Weaknesses’.

**Limitations:**

The limitations are adequately discussed.

---

> ### Author Rebuttal · Authors · 2023-08-08
>
> We are grateful for your positive comments about the implications of our work.
>
> > It is unclear why the accuracy does not go down in the empirical results presented.
>
> The accuracy does go down slightly (not quite visible in original Figure 4a, but definitely visible in original Figure 5e.
>
> > If the input signal is not being integrated in the case of the refractory period, then that part of the signal is forever lost. This should naturally lead to a decrease in accuracy by the network. Please elaborate why this is not happening.
>
> This is happening. If the restriction on the number of spikes becomes too harsh (e.g. due to a large ARP), the performance of the network will decline. This is the reason why multi-spike SNNs tend to outperform single-spike SNNs (in which neurons can spike at most once; Eshraghian et al., 2021). However, during training, our method is still able to learn from the entire input signal (as our implementation permits gradients to flow at every time step in the network, even when the neurons are in a refractory state). Thus, our method can learn which parts of the signal are most important (and which less so) and hence the refractory period only causes a small reduction in accuracy unless it is very large.
>
> > Is this something specific to the tasks that you are performing, and how is the accuracy impacted in general?
>
> We found we could obtain good accuracies on the machine learning benchmarks using even large ARPs (tens of milliseconds; original Figure 4a), however we had to use smaller ARPs for the neural-fitting task (less than ten milliseconds - which is expected, as you cannot fit arbitrary spike trains with a restricted number of spikes; original Figure 5e). In general, as the ARP tends to infinity, the accuracy of the network would tend to the performance of a single-spike SNN.
>
> > Simple simulations with a couple of neurons may help here, while varying the ARP. This may also give some intuition as to how to pick this number to minimize loss of accuracy while still obtaining speedups.
>
> We have run additional experiments increasing the ARP on the machine learning benchmarks and the neural-fits, and - confirming your hunch - showcase more of a performance drop in accuracy (see new Figure 4 and 5). We hope this additional information will be helpful to users in balancing the accuracy/speedup tradeoff at large ARPs.

---

> > ### Comment · Reviewer_x3Zh · 2023-08-14
> > **Response to rebuttal**
> >
> > Thanks for the clarification! I am happy to increase my rating based on the authors' new experiments and clarifications. Useful paper!

---

### Official Review · Reviewer_gA8o · 2023-07-28

**Soundness:** 4 excellent
**Presentation:** 3 good
**Contribution:** 4 excellent
**Rating:** 7
**Confidence:** 4

**Summary:**

Spiking neural networks can be computationally expensive to implement because of their history dependence.  However, spikes can't occur at any time whatsoever; instead, after spiking, there is a refractory period during which they cannot spike again. While this period is ~1-2ms, to attain precise dynamical inferences the typical time step is ~0.1ms.  The authors leverage the former fact, creating a block design, each block length equaling 1 ARP, to substantially reduce the number of required sequential simulation steps while maintaining generally high performance on both ML and neural-fitting tasks.

**Strengths:**

This appears to be an important and very simple pragmatic algorithm that can benefit the field by greatly reducing SNN simulation times.  The general presentation is clear.

**Weaknesses:**

Minor:

Fig 5:

b) A timescale bar is needed, as is the ARP used.  Given that there is very little performance change with DT=0.1ms, ARP=8ms (e), it would be helpful to visualize this here as well

d-f) Bar graphs that do not go to zero and have no broken axis can lead to confusion.  One or the other should be done.  I also would advocate for reversing the ordering: f, e, d.  f and e are the most compelling graphs, and f (left) provides an excellent overview comparison to begin with.  d, while relevant, is also not surprising and could be better left as a final, expected finding.

Wording:

Lines 266 and 270, "computational neuroscientists" and "biologists."  I understand where the authors are coming from; however, both are of interest to computational neuroscientists (who might do theory, computational studies, and/or data analysis), while "biologists" is overly broad.  I would suggest simply indicating that both are of interest to computational and experimental neuroscientists, or to change "biologists" to "computational and experimental neuroscientists".

Line 290:

I'm not sure this is a fair characterization of your impressive findings in Fig 5e.  The larger ARPs had little performance effect in that particular figure, while larger DTs had a large effect.

**Questions:**



**Limitations:**

Y

---

> ### Author Rebuttal · Authors · 2023-08-08
>
> We thank you for your positive comments about the strengths of our paper and suggested edits to improve our work.
>
> > Fig 5b): A timescale bar is needed, as is the ARP used. Given that there is very little performance change with DT=0.1ms, ARP=8ms (e), it would be helpful to visualize this here as well
>
> We have added a timescale bar, the ARP used, and included your requested visualization (see updated Figure 5b).
>
> > Fig 5d-f: Bar graphs that do not go to zero and have no broken axis can lead to confusion. One or the other should be done. I also would advocate for reversing the ordering: f, e, d. f and e are the most compelling graphs, and f (left) provides an excellent overview comparison to begin with. d, while relevant, is also not surprising and could be better left as a final, expected finding.
>
> We have modified the bar graphs to go to zero and have rearranged the graphs as suggested (and have updated the text accordingly). See updated Figure 5.
>
> > [Wording:] Lines 266 and 270, "computational neuroscientists" and "biologists." I understand where the authors are coming from; however, both are of interest to computational neuroscientists (who might do theory, computational studies, and/or data analysis), while "biologists" is overly broad. I would suggest simply indicating that both are of interest to computational and experimental neuroscientists, or to change "biologists" to "computational and experimental neuroscientists".
>
> We agree and have changed “biologists” to “computational and experimental neuroscientists”.
>
> > [Wording:] Line 290: I'm not sure this is a fair characterization of your impressive findings in Fig 5e. The larger ARPs had little performance effect in that particular figure, while larger DTs had a large effect.
>
> Thank you. To still highlight the applicability of our method (in the context of its limitations), we have reworded line 290 to:
> “However, we found a beneficial trade-off, by using large non-biological ARPs on the artificial spiking classification datasets and smaller physiological ARPs for the neural fits (although we found larger values to also perform reasonably well)”

---

> > ### Comment · Reviewer_gA8o · 2023-08-21
> >
> > Thank you for addressing the minor comments I had.  My score remains a 7.

---

### Author Rebuttal · Authors · 2023-08-08

Dear reviewers,

Thank you for taking the time to review our paper.  As requested, we have run additional experiments, and have appropriately updated Figures 4 and 5, which we have uploaded in the attached single-page PDF. Furthermore, we have included a zoomed-in Supplementary plot of Figure 5b of the neural traces. Lastly - as requested by reviewer 9z9h - we have made all the code available (which we have shared with the AC as set out in the NeurIPS guidelines).

---

### Author Response · Authors · 2023-08-15
**Reminder to AC to share code**

Dear AC, reviewer 9z9h has not been able to access the provided code. Could we please request for this to be made available? Thank you.

---

### Decision · Program_Chairs · 2023-09-21

**Decision:**

Accept (poster)

**Comment:**

The authors propose a method to significantly increase the simulation time for spiking neural networks (SNNs) based on the adaptive leaky-integrate-and-fire (ALIF) neuron model on GPUs. To do so, they utilize the refractory period of neurons in which no spike of the neuron can appear. The method is evaluated on standard SNN machine learning benchmarks and biologically motivated simulations.

All reviewers agree that the manuscript is well-written and that the manuscript presents an interesting contribution to the SNN simulation community. The achieved simulation and training speedups are impressive (up to 50x) with little influence on network performance.
The initial submission did not compare to other SoTA simulations. This was performed in the rebuttal phase, with convincing results.

One weakness of the manuscript is that the originality is somewhat limited as the general idea has been implemented previously for SNN simulation. One advance is that the proposed approach is used also for training and on GPUs.

The evaluations somewhat and the manuscript is close-to-borderline. Nevertheless, the very good results suggest that the work can be accepted for NeurIPS.